# Senescent-like macrophages mediate angiogenesis for endplate sclerosis via IL-10 secretion in male mice

Yonggang Fan[1], Weixin Zhang[2], Xiusheng Huang[1], Mingzhe Fan[1], Chenhao Shi[1], Lantian Zhao[1], Guofu Pi[1], Huafeng Zhang[1] & Shuangfei Ni [1] ✉

Endplate sclerosis is a notable aspect of spine degeneration or aging, but the mechanisms remain unclear. Here, we report that senescent macrophages accumulate in the sclerotic endplates of lumbar spine instability (LSI) or aging male mouse model. Specifically, knockout of cdkn2a (p16) in macrophages abrogates LSI or aging-induced angiogenesis and sclerosis in the endplates. Furthermore, both in vivo and in vitro studies indicate that IL-10 is the primary elevated cytokine of senescence-related secretory phenotype (SASP). Mechanistically, IL-10 increases pSTAT3 in endothelial cells, leading to pSTAT3 directly binding to the promoters of Vegfa, Mmp2, and Pdgfb to encourage their production, resulting in angiogenesis. This study provides information on understanding the link between immune senescence and endplate sclerosis, which might be useful for therapeutic approaches.

Low back pain (LBP) is a frequent and significant cause of disability worldwide[1,2]. About 80% of people will experience LBP at some point in their lives, making it a common and persistent symptom[3]. This ongoing pain is linked to the emergence of numerous physical and psychosocial impairments, which led to a huge financial burden[4,5].

The cartilage endplate, situated between the nucleus pulposus and the bony vertebral body, is vulnerable to degeneration and damage due to being subject to high shear stresses[6]. The cartilage endplate develops progressive sclerosis with aging or in response to other degenerative conditions[7]. Magnetic resonance imaging (MRI) findings of bone marrow lesions within endplates known as "Modic changes (MC)" are predictive of high LBP specificity[8–10]. In our previous work, we identified endplate sclerosis in lumbar spine instability (LSI) and aging model, and spinal pain was brought on by sensory innervation in the porous area of sclerotic endplates[11]. However, the mechanism underlying LSI or aging-induced endplate sclerosis is yet unclear.

Given the emerging involvement of cellular senescence in bone during aging, we contrive to investigate whether cellular senescence induces endplate sclerosis. Cellular senescence is characterized by irreversible, sustained cell cycle arrest. It occurs not just in certain age-related disorders but also throughout the life cycle, including embryonic and adult life[12,13]. Senescent cells can interfere with tissue homeostasis and function by secreting senescence-related secretory phenotype (SASP) composed of inflammatory cytokines, growth factors, and proteases[14]. In the skeletal system, a growing emphasis has been given to the role that cellular senescence exerts in the onset of osteoarthritis[15], osteoporosis[16], skeletal growth[17], and intervertebral disc nucleus pulposus degeneration[18]. The senolytic drugs could delay age-related intervertebral disc degeneration[19]. However, uncertainty exists over the part that cellular senescence plays in the development of endplate degeneration and sclerosis.

Macrophages are intrinsic immune cells in the body[20]. In response to external stimuli such as injury and mechanical load, macrophages rapidly react to convergence and undergo phenotypic and functional changes. This aids in regulating the local immune response and maintaining metabolic homeostasis[21–23]. Macrophages can undergo considerable phenotypic alterations upon senescence, shifting from a phenotype that hinders angiogenesis to a phenotype that promotes it[24–26]. Blood vessels are essential for bone development and neogenesis in the mammalian skeletal system[27,28]. The term "angiogenesis osteogenesis coupling" refers to the strong geographical and temporal

[1]Department of Orthopaedics, 1st Affiliated Hospital of Zhengzhou University, Zhengzhou 450000, PR China. [2]Zhejiang Chinese Medicine University, Hangzhou 310053, PR China. ✉e-mail: nishuangfei@yeah.net

relationship between osteogenesis and angiogenesis[29]. The osteogenesis-related capillary subtype known as type H vessels, which is characterized by high expression of CD31 and Endomucin (CD31^hi Emcn^hi), was recently identified[30–32]. However, angiogenesis is not always beneficial, for instance, angiogenesis in tumor tissue[33], angiogenesis in retinopathy[34], and angiogenesis in subchondral bone in osteoarthritis[35], all of which hasten the onset and progression of the illness. Under normal conditions, the endplate is an avascular cartilaginous tissue that relies on its capacity to diffuse nutrients from neighboring blood vessels to both itself and the nucleus pulposus[36]. The cartilage endplate gradually becomes harder as they age and degenerate, transiting from avascular cartilaginous tissue to osteogenic tissue containing blood vessels[37]. This suggests that senescent-like macrophages may be involved in endplate sclerosis-related angiogenesis.

In this study, we sought to demonstrate that senescent-like macrophages influence immune-vascular communication by secreting the pro-angiogenic cytokine IL-10, which causes endplate sclerosis by affecting the Signal Transducer and Activator of Transcription 3 (STAT3) signaling pathway. According to our data, sclerotic endplate cavities contained more F4/80⁺P16⁺ senescent-like macrophages. Endplate sclerosis was exacerbated by senescent-like macrophages in porous endplates that promoted CD31⁺Emcn⁺ angiogenesis. Endplate sclerosis was greatly reduced by eliminating senescent-like macrophages or inhibiting IL-10.

## Results

### CD31⁺Emcn⁺ vessels couple with endplate sclerosis in LSI and aged mice

A unique capillary subtype known as CD31^hi Emcn^hi type-H vessels couple angiogenesis with osteogenesis[32]. We showed previously that LSI or aging-induced spine degeneration could initiate the sclerosis of endplates[11]. We analyzed the caudal endplates of L4/5 to determine whether CD31⁺Emcn⁺ blood vessels were involved in the sclerosis of endplates in LSI or aged mice. Immunofluorescent staining demonstrated that CD31⁺Emcn⁺ blood vessels were significantly growing into the porous area of sclerotic endplates at 4 weeks and 8 weeks after LSI surgery (Fig. 1a, c) or in 20-month-old mice (Fig. 1b, d). While no evident sprouting of CD31⁺Emcn⁺ blood vessels was observed in the cartilaginous endplates of sham surgery mice (Fig. 1a, c) or 3-month-old mice (Fig. 1b, d). Then, the porosity of endplates increased significantly in LSI surgery mice (Fig. 1e, g) or aged mice (Fig. 1f, h), as evidenced by three-dimensional micro-computed tomography (micro CT) analysis. Safranin O and fast green staining demonstrated that the green-stained bone matrix surrounded the cavities in endplates of LSI surgery mice (Fig. 1i) or aged mice (Fig. 1j), suggesting endochondral ossification. The endplate scores for pathological evaluation were significantly higher in LSI surgery mice (Fig. 1k) or aged mice (Fig. 1l). Importantly, the number of osterix (Osx)⁺ osteoblast progenitors and osteocalcin (OCN)⁺ osteoblasts also increased significantly in the endplates at 4 weeks and 8 weeks after LSI surgery (Fig. 1m, o, p) or in 20-month-old mice (Fig. 1n, q, r) relative to control mice. These indicate that the endplate undergoes more osteogenesis during degeneration.

To further validate the coupling of angiogenesis and endplate sclerosis, a pan-VEGF receptor tyrosine kinase inhibitor, SU5416 was administered to LSI surgery mice. Immunofluorescent staining showed that LSI-induced sprouting of CD31⁺Emcn⁺ blood vessels in endplates was significantly inhibited by SU5416 treatment (Supplementary Fig. 1a, b). Notably, SU5416 treatment significantly halted the endplate sclerosis in LSI surgery mice, as determined by micro CT analysis (Supplementary Fig. 1c, d) and safranin O and fast green staining (Supplementary Fig. 1e, f). Taken together, these results suggest that endplate sclerosis during degeneration potentially depends on CD31⁺Emcn⁺ blood vessels.

### Senescent cells accumulate in the sclerotic endplates in LSI mice and aged mice

To clarify the potential involvement of cellular senescence in endplate sclerosis, we first conducted SA-βGal staining and found a significant increase in the number of SA-βGal⁺ cells in the endplates at 4 weeks and 8 weeks after LSI surgery (Fig. 2a, b). Here, we employed a murine senescence reporter strain cyclin dependent kinase inhibitor 2A (Cdkn2a) (p16)^tdTom, in which tandem-dimer Tomato (tdTom) was inserted into exon 2 of the Cdkn2a locus to enable the identification of Cdkn2a activated cells (tdTom⁺) at the single-cell level. Immunofluorescent staining revealed that the number of tdTom⁺ cells increased significantly at 4 weeks and 8 weeks after LSI surgery relative to sham surgery mice (Fig. 2a, c), indicating high-level activation of the Cdkn2a promoter. Additionally, the majority of the cartilaginous endplate cells in sham surgery mice are positive for nuclear localization of HMGB1, whereas the number of HMGB1⁺ cells decreased significantly with overall relocalization of HMGB1 in the endplates at 4 weeks and 8 weeks after LSI surgery, as indicated by immunofluorescent staining (Fig. 2d, e). To further demonstrate the presence of senescence, real-time quantitative PCR (qRT-PCR) was performed on the Cdkn2a (p16), Trp53 (p53), and Cdkn1a (p21) genes, as well as the typical SASP Il1b (IL-1β) and Il6 (IL-6). The mRNA expressions of them were significantly higher in LSI surgery mice relative to sham surgery mice (Fig. 2f, g). These findings imply LSI-induced cellular senescence in the endplates.

In parallel, we assessed the cellular senescence in the endplates during aging. Similarly, SA-βGal staining or immunofluorescent staining showed that the number of SA-βGal⁺ cells or tdTom⁺ cells in the endplates was significantly higher in 20-month-old mice relative to 3-month-old mice (Fig. 2h–j). Moreover, the reduced immunofluorescent intensity and relocalization of HMGB1 were also observed in the endplates of 20-month-old mice (Fig. 2k, l). Aging produced higher mRNA expression of Cdkn2a (p16), Trp53 (p53), Cdkn1a (p21), Il1b (IL-1β), and Il6 (IL-6) according to qRT-PCR (Fig. 2m, n). Together, these data indicate that the endplate also experiences cellular senescence as they age.

### The sclerosis of endplates is delayed by the clearance of senescent cells in LSI and aged mice

Based on the above experimental findings, it is logical to postulate that cellular senescence is potentially associated with angiogenesis and osteogenesis in the endplates during degeneration. Intermittent delivery of Dasatinib plus Quercetin (D + Q) is a typical and recognized senolytic cocktail for selective elimination of senescent cells. Here, D + Q or Dimethyl sulfoxide (DMSO) was administered to LSI surgery mice (Fig. 3a). D + Q treatment significantly decreased the number of SA-βGal⁺ cells in the endplates at 4 weeks (Supplementary Fig. 2a, b) and 8 weeks (Fig. 3b, c) after LSI surgery relative to that with DMSO treatment, according to SA-βGal staining. Immunofluorescent staining demonstrated that the number of tdTom⁺ cells was significantly decreased by D + Q treatment in Cdkn2a (p16)^tdTom mice (Fig. 3d, e). Further, co-immunofluorescent staining of CD31 and Emcn demonstrated that D + Q treatment abolished LSI-induced angiogenesis in the endplates at 4 weeks (Supplementary Fig. 2c, d) and 8 weeks (Fig. 3f, g). Micro CT analysis showed that LSI-induced sclerosis of endplates was significantly abrogated by D + Q treatment (Fig. 3h, i). Moreover, Safranin O and fast green staining demonstrated that the endplate sclerosis was significantly delayed in mice receiving D + Q treatment at 4 weeks (Supplementary Fig. 2e, f) and 8 weeks (Fig. 3j, k) after LSI surgery, as indicated by less porous area and significantly lower endplate scores. The presence of Osx⁺ osteoblast progenitors in the endplates of LSI surgery mice was also impeded by D + Q treatment in immunohistochemical staining (Fig. 3l, m). Additionally, the western blot and qRT-PCR analysis demonstrated that LSI surgery significantly induced the expression of F4/80, CD31, Osx, P16, and γH2A.X in the

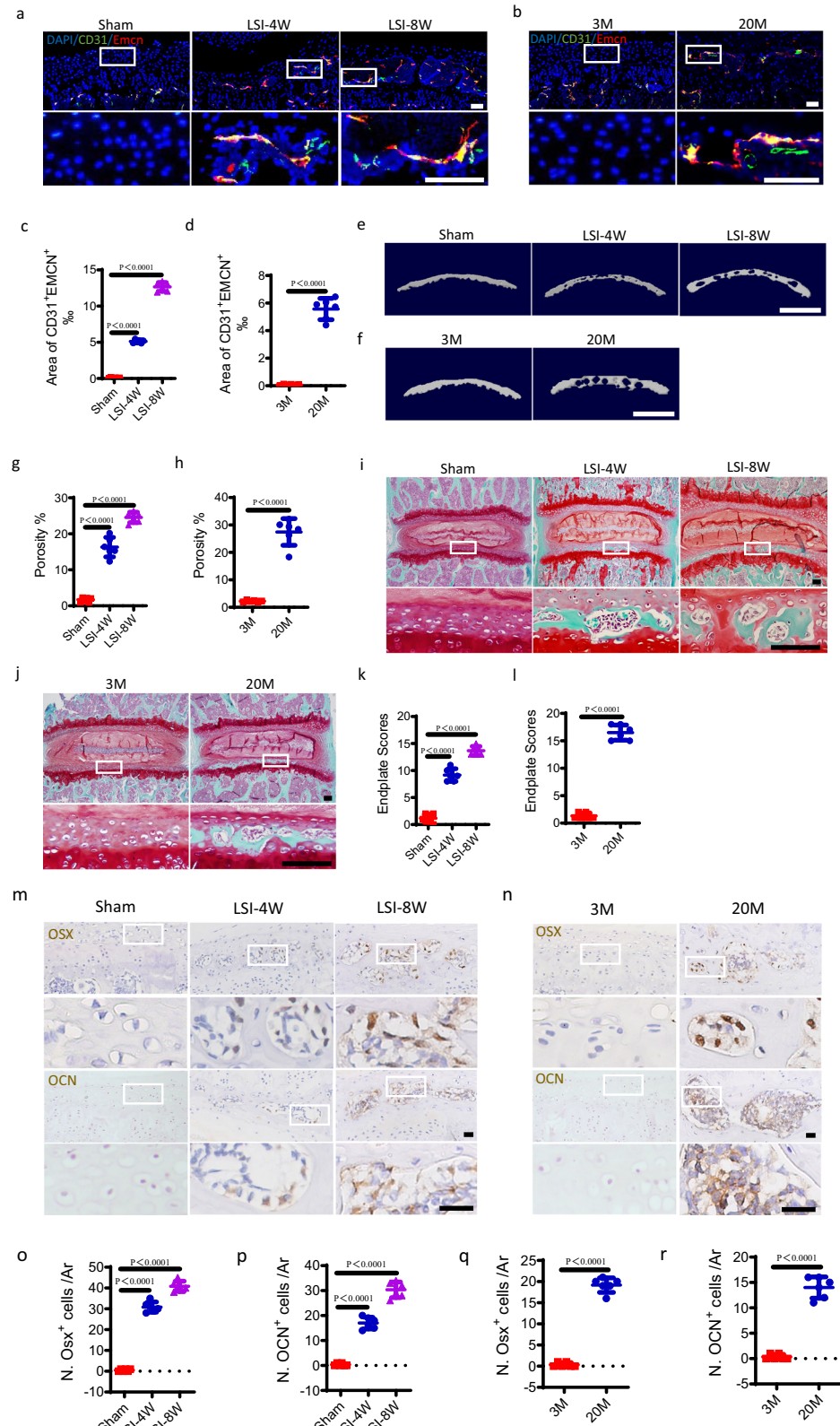

endplates, while, the D + Q treatment prevented the LSI-induced increase of them in the endplates (Supplementary Fig. 3a–c).

To further support these findings, D + Q was also administrated to aged mice (Fig. 4a). D + Q treatment significantly inhibited the aging-induced rise in the number of SA-βGal⁺ cells in the endplates, as indicated by SA-βGal staining (Fig. 4b, c). Remarkably, the number of CD31⁺Emcn⁺ blood vessels decreased significantly in the endplates in 20-month-old mice with D + Q treatment relative to that with DMSO treatment (Fig. 4d, e). Consistently, micro CT analysis (Fig. 4f, g) and Safranin O and fast green staining (Fig. 4h, i) demonstrated that D + Q treatment partially impeded the sclerosis of endplates in 20-month-old mice. The number of Osx⁺ osteoblast progenitors in the endplates of 20-month-old mice also decreased significantly by D + Q treatment (Fig. 4j, k). Taken together, these results imply that the senolytic

**Fig. 1 | CD31⁺Emcn⁺ vessels appear in sclerotic endplates in LSI and aged mice.** Representative immunofluorescent images of CD31⁺ (green), Emcn⁺ (red) cells and DAPI (blue) staining of nuclei in the caudal endplates of L4/5 at 4 and 8 weeks after LSI surgery or sham surgery (**a**) or in 3-month-old or 20-month-old mice (**b**). Scale bars, 50 μm. **c, d** Permillage of CD31⁺Emcn⁺ area of **a** and **b**. Representative three-dimensional μCT images of the caudal endplates of L4/5 (coronal view) at 4 and 8 weeks after LSI surgery or sham surgery mice (**e**) or in 3-month-old or 20-month-old mice (**f**). Scale bars, 500 μm. **g, h** Quantitative analysis of the total porosity of **e** and **f**. Representative images of safranin O and fast green staining of endplate sections at 4 and 8 weeks after LSI or sham surgery (**i**) or in 3-month-old or 20-

month-old mice (**j**), proteoglycan (red) and cavities (green). Scale bars, 50 μm. **k, l** Endplate scores as an indication of endplate degeneration based on safranin O and fast green staining. Representative immunohistochemical images of Osterix (Osx, top) or Osteocalcin (Ocn, bottom) in the endplates at 4 weeks or 8 weeks after LSI or sham surgery (**m**) or in 3-month-old or 20-month-old mice (**n**). Scale bars, 50 μm. **o–r** Quantitative analysis of the number of Osx⁺ cells or Ocn⁺ cells of **m** and **n**. $n = 6$ per group. Data are represented as means ± standard deviations, as determined by two-tailed Student's $t$ test or One-way ANOVA. Source data are provided as a Source Data file.

---

cocktail, D + Q could inhibit endplate angiogenesis and sclerosis induced by LSI surgery or aging.

## Abundant macrophages undergo senescence in the endplates of LSI mice and aged mice

To identify the senescent cells in degenerative endplates, we initially conducted immunofluorescent staining in Cdkn2a (p16)^tdTom mice. The co-staining of F4/80 with tdTom showed that the number of F4/80⁺tdTom⁺ cells in the endplates increased significantly at 4 weeks (Fig. 5a, c) and 8 weeks after LSI surgery (Supplementary Fig. 4a, b) or 20-month-old mice (Fig. 5b, d) relative to control mice. Then, we isolated the cells in the endplates of Cdkn2a (p16)^tdTom mice undergoing sham or LSI surgery to perform the flow cytometry analysis. Consistently, the results showed that the number of F4/80⁺tdTom⁺ cells increased significantly in response to LSI surgery (Fig. 5e, f and Supplementary Fig. 5). To investigate when senescent-like macrophages infiltrate the endplates, we conducted co-staining of P16 with F4/80 in the endplates from mice of different ages. The results showed that there were few P16⁺F4/80⁺ cells in the endplates of 3-month-old mice, while in 6-month-old mice, the infiltrated F4/80⁺ cells began to appear. Interestingly, we observed a high proportion of co-localization of P16 with F4/80 in the endplates of 6-month-old mice. Subsequently, the amount of P16⁺F4/80⁺ cells increased in the endplates of 12-month-old mice and remained at a high level in the endplates of 16, and 20-month-old mice (Supplementary Fig. 6a, b). Together, these data indicate that macrophages are characterized by high expression of p16 upon infiltration into the endplates. To validate these F4/80⁺ cells are indeed macrophages, we conducted the co-immunostaining of F4/80 with CD68 or CD11b in the LSI surgery mice and aged mice. The results demonstrated that F4/80 was mostly co-localized with CD68 and CD11b in the sclerotic endplates (Supplementary Fig. 7a). To clarify the sub-type of the senescent-like macrophages, we conducted the co-immunostaining of P16 with M2 marker, CD206 or M1 marker, iNOS. The data revealed that the majority of the P16-positive cells expressed the M2 polarization marker CD206, as opposed to the M1 polarization marker iNOS in the endplates of LSI surgery mice and aged mice (Supplementary Fig. 7b, c). This suggests that most senescent-like macrophages tend to exhibit the M2 marker.

Subsequently, we tested the sensitive molecular markers for tracking DNA damage, including senescence-associated distension of satellites (SADS) and γH2A.X foci, which are characteristic features of senescent cells. The presence of SADS in F4/80⁺ cells increased significantly in the endplates of LSI surgery mice or 20-month-old mice relative to control mice, as evidenced by immunofluorescence in situ hybridization (FISH) combined with immunofluorescent staining (Fig. 5g, h and Supplementary Fig. 8). Consistently, immunofluorescent staining also demonstrated that the degenerative endplates contained more γH2A.X⁺F4/80⁺ cells in LSI surgery mice or 20-month-old mice relative to control mice, suggesting DNA damage in F4/80⁺ cells in sclerotic endplates (Fig. 5i, j). These findings indicate that F4/80⁺ macrophages in degenerative endplates display profound induction of p16 expression as well as specific cellular senescence markers (SADS and γH2A.X). Moreover, western blot analysis demonstrated that the protein level of Bcl2 increased, while the protein level

of Bax decreased in H₂O₂-treated bone marrow-derived macrophages (BMDMs) (Supplementary Fig. 9a, b), indicating the resistance to apoptosis in senescent-like macrophages. In addition, co-staining of F4/80 with CD31 revealed that F4/80⁺ cells were situated adjacently to CD31⁺ endothelial cells, suggesting the potential correlation of macrophages with angiogenesis in the endplates of LSI surgery mice (Fig. 5k) or 20-month-old aged mice (Fig. 5l). Collectively, these results suggest that macrophage experienced senescence in degenerative endplates.

## Knockout of cdkn2a (p16) in Lyz2⁺ cells abrogates endplate sclerosis

To further elucidate the function of senescent-like macrophages in the development of endplate sclerosis, we bred Lyz2-Cre mice with Cdkn2a (p16) floxed mice to generate Lyz2^cre; Cdkn2a^flox/flox mice (named Cdkn2a^ΔLyz2) to exclusively knockout Cdkn2a in Lyz2⁺ macrophages (Fig. 6a). The SADS detection showed that the conditional knockout of Cdkn2a in Lyz2⁺ cells or D + Q treatment inhibited the presence of SADS in the F4/80⁺ macrophages in the LSI surgery mice (Supplementary fig. 10). The sprouting of CD31⁺Emcn⁺ blood vessels in the endplates was significantly inhibited at 8 weeks after LSI surgery in Cdkn2a^ΔLyz2 mice relative to their age-matched wild-type (WT) littermates (Cdkn2a^f/f mice), as determined by immunofluorescent staining (Fig. 6b, c). Micro CT analysis demonstrated that the sclerosis of endplates was significantly delayed after LSI surgery in Cdkn2a^ΔLyz2 mice relative to Cdkn2a^f/f mice, as shown by a reduction in endplate porosity (Fig. 6d, e). This finding was further verified by Safranin O and fast green staining, with less porous areas and significantly lower endplate scores (Fig. 6f, g). Additionally, immunohistochemical staining demonstrated that the number of Osx⁺ osteoblast progenitors decreased significantly at 8 weeks after LSI surgery in Cdkn2a^ΔLyz2 mice relative to Cdkn2a^f/f mice (Fig. 6h, i). Tartrate-resistant acid phosphatase (Trap) staining showed that the number of Trap⁺ osteoclasts after LSI surgery was lower in the endplates of Cdkn2a^ΔLyz2 mice relative to Cdkn2a^f/f mice (Supplementary Fig. 11). Taken together, these results indicate that deletion of Cdkn2a in Lyz2⁺ cells halts LSI surgery-induced angiogenesis and osteogenesis in vertebral endplates.

We previously reported that the instability or aging could induce CGRP⁺ nociceptive sensory innervation in the endplates to cause spine hypersensitivity[11]. Notably, in this study, immunostaining showed that the density of CGRP⁺ sensory nerves was decreased significantly in Cdkn2a^ΔLyz2 mice relative to Cdkn2a^f/f mice (Fig. 6j, k). Although the density of CGRP⁺ sensory nerve fibers increased slightly at 8 weeks after LSI surgery in Cdkn2a^ΔLyz2 mice, it was still significantly lower than that of Cdkn2a^f/f mice (Fig. 6j, k). Additionally, the LSI surgery-induced increased density of CGRP⁺ sensory nerves in the endplates was diminished by D + Q treatment (Supplementary Fig. 12).

## Deletion of macrophages inhibits LSI- or aging-induced endplate sclerosis

Considering the high portion of senescent-like macrophages in degenerative endplates, we next examined whether the removal of macrophages can prevent angiogenesis and sclerosis in the endplates during degeneration. Lyz2-cre mice were bred with B6-iDTR mice to

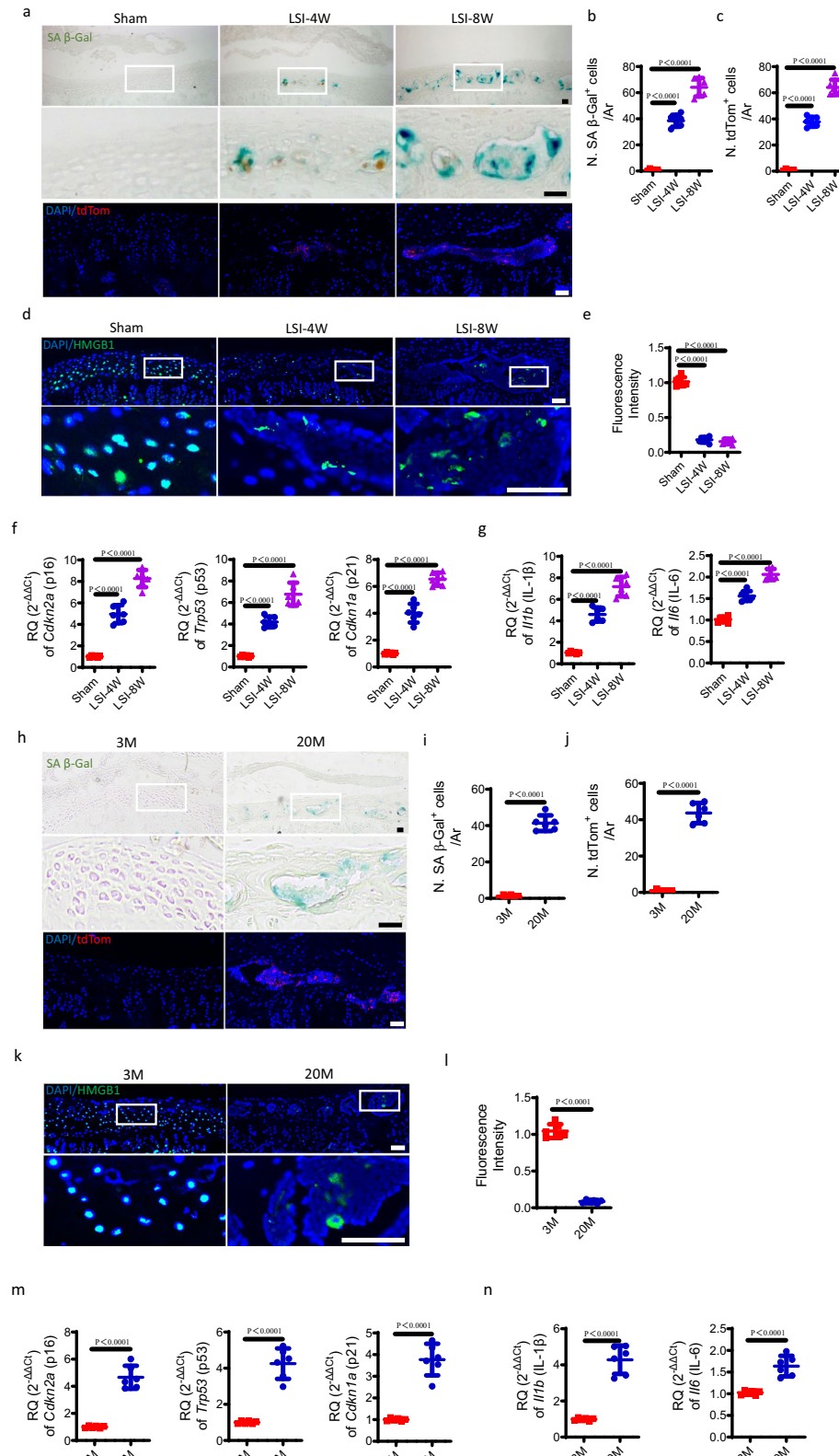

generate Lyz2cre; iDTR mice (named iDTR△Lyz2) to delete Lyz2+ cells specifically by diphtheria toxin (DT) injection. The efficacy of DT injection was confirmed by the decreased number of F4/80+ cells in the endplates at 8 weeks after LSI surgery in iDTR△Lyz2 mice, as indicated by immunofluorescent staining (Fig. 7a). DT injection significantly suppressed the growth of CD31+Emcn+ blood vessels in the endplates of iDTR△Lyz2 mice at 8 weeks after LSI surgery (Fig. 7b, c). As expected, the

micro CT analysis revealed that the sclerosis of endplates was also significantly postponed in LSI surgery iDTR△Lyz2 mice by DT injection, as shown by the decrease in endplate porosity (Fig. 7d, e). Furthermore, Safranin O and fast green staining validated the delay of endplate sclerosis by DT injection in iDTR△Lyz2 mice at 4 weeks (Supplementary Fig. 13a, b) and 8 weeks (Fig. 7f, g) after LSI surgery. The DT injection also significantly reduced the LSI-induced increase of Osx+ osteoblast

**Fig. 2 | Senescent cells accumulate in the sclerotic endplates of LSI mice and aged mice. a** Representative images of coronal mouse caudal endplate sections of L4/5 stained for SA-βGal (green) at 4 and 8 weeks after LSI or sham surgery (top). Representative immunofluorescent images of tdTom⁺ cells (red) and DAPI (blue) staining of nuclei at 4 and 8 weeks after LSI or sham surgery in p16^tdTom mice (bottom). Quantitative analysis of the number of SA-βGal⁺ cells (**b**) or tdTom⁺ cells (**c**) of **a. d** Representative immunofluorescent images of HMGB1⁺ cells (green) and DAPI (blue) staining of nuclei in endplates at 4 and 8 weeks after LSI or sham surgery. **e** Quantified fluorescence intensity of HMGB1⁺ cells of **d. f, g** Relative expression of senescence pathway genes, Cdkn2a (p16), Trp53 (p53), and Cdkn1a (p21) (**f**) or typical SASP, Il1b (IL-1β) and Il6 (IL-6) in the endplates at 4 and 8 weeks after LSI or sham surgery. **h** Representative images of SA-βGal (green) staining (top) or tdTom⁺ cells (red) and DAPI (blue) staining of nuclei (bottom) in 3-month-old or 20-month-old mice. **i, j** Quantitative analysis of **h. k** Representative immunofluorescent images of HMGB1⁺ cells (green) and DAPI (blue) staining of nuclei in 3-month-old or 20-month-old mice. **l** Quantified fluorescence intensity of **k.** Relative expression of senescence pathway genes (**m**) or typical SASP (**n**) in the endplates of 3-month-old or 20-month-old mice. Scale bars, 50 μm. *n* = 6 per group. Data are represented as means ± standard deviations, as determined by two-tailed Student's *t* test or One-way ANOVA. Source data are provided as a Source Data file.

progenitors in the endplates of iDTR^ΔLyz2 mice, as determined by immunohistochemical staining (Fig. 7h, i).

The administration of clodronate liposomes (CL) is efficient for macrophage depletion[38,39]. To further validate the above results in aged mice, CL was injected to delete macrophages. The effective depletion was confirmed by immunofluorescent staining that showed a reduced number of F4/80⁺ cells in the endplates of 20-month-old mice by CL injection (Fig. 8a). CL administration significantly decreased the area of CD31⁺Emcn⁺ blood vessels in the endplates of 20-month-old mice relative to vehicle-treated aged mice, as evidenced by immunofluorescent staining (Fig. 8b, c). Micro CT examination demonstrated that the endplate sclerosis was partially rescued by CL treatment in 20-month-old mice (Fig. 8d, e). Consistently, Safranin O and fast green staining showed decreased sclerosis of endplates in 20-month-old mice with CL administration (Fig. 8f, g). The presence of Osx⁺ osteoblast progenitors in the endplates was also inhibited by CL treatment in 20-month-old mice (Fig. 8h, i), as determined by immunohistochemical staining. These results collectively imply that the macrophage depletion solution, CL could prevent the angiogenesis and sclerosis of endplates induced by aging. Additionally, the genetic or pharmacological clearance of macrophages in LSI surgery mice or aged mice could decrease the number of Trap⁺ osteoclasts in the endplates (Supplementary Fig. 14).

## IL-10 secreted by senescent-like macrophages contributes to angiogenesis in degenerative endplates

To elucidate the potential mechanism of senescent-like macrophage-mediated angiogenesis in the endplates, we first analyzed target genes from immune system functions that fall into 9 classes: Cell Surface Receptors, Stress Response, Oxidoreductases, Proteases, Transcription Factors, Signal Transduction, Cytokines & Cytokine Receptors, Chemokines & Chemokine Receptors, and Cell Cycle & Protein Kinases using a Mouse Immune Response Plate. We found that messenger ribonucleic acid (mRNA) levels of the following genes were the most modified in the endplates of 20-month-old mice relative to 3-month-old mice: Il10 (IL-10), Tnf (TNF), Il1b (IL-1β), Nfkb2 (NFκB2), Ccl2 (CCL2), Ccr2 (CCR2), Lrp2 (LRP2), Ikbkb (IKK2), Il18 (IL-18), Csf3 (G-CSF), Nfatc3 (NFAT4), Bcl2l1 (BCL2L), Cd38 (CD38), Ace (CD143), Icam1 (CD54), and Fn1 (FN-1) (Fig. 9a). Among them, IL-10 was the most elevated gene in the endplates of 20-month-old mice (Fig. 9a). The increased protein level of IL-10 in the endplates of aged mice was validated by enzyme-linked immunoabsorbent assay (ELISA) (Fig. 9b). Immunofluorescent staining further demonstrated that tdTom⁺F4/80⁺ senescent-like macrophages were the main source of IL-10 secretion in the endplates at 8 weeks after LSI surgery (Fig. 9c). To validate the increased release of IL-10 from senescent-like macrophages, the primary BMDMs were isolated for in vitro experiments (Fig. 9d). The senescence of BMDMs was induced by H₂O₂, which was confirmed by SA-βGal staining (Fig. 9e). We then examined the expression of several altered inflammatory cytokines and Cdkn2a (P16) in vitro using qRT-PCR. The results confirmed the secretion of SASP and the significant increase of IL-10 and P16 mRNA levels in H₂O₂-induced senescent BMDMs (Fig. 9f). Importantly, ELISA showed that the conditioned medium (CM) of H₂O₂-treated BMDMs contained much more IL-10

than the CM of naive BMDMs (Fig. 9g). While the 2A5, an anti-IL-10 neutralizing antibody, inhibited H₂O₂-induced IL-10 release (Fig. 9g). Interestingly, co-immunofluorescence of IL-10 with CD206 or iNOS showed that the cells expressing IL-10 mainly co-localized with CD206⁺ cells rather than iNOS⁺ cell in the endplates of LSI surgery mice and aged mice (Supplementary Fig. 15). More importantly, immunofluorescence of IL-10 with F4/80 or P16 demonstrated that the number of double-positive cells in the endplates of LSI group was significantly diminished in the Cdkn2a^ΔLyz2 mice relative to Cdkn2a^f/f mice (supplementary Fig. 16). QRT-PCR demonstrated that the increased mRNA expressions of Il10 (IL-10), Il1b (IL-1β), and Tnfα (TNF-α) induced by LSI surgery were decreased by macrophage deletion in iDTR^ΔLyz2 mice with DT injection (supplementary Fig. 17).

Subsequently, we examined the effect of different CMs from BMDMs on the tube formation of endothelial cells. In comparison to the CM of naive BMDMs, we found that the CM of H₂O₂-treated BMDMs significantly boosted the tube formation of endothelial cells (Fig. 9h, i). While, the 2A5 prevented the tube formation of endothelial cells induced by CM of H₂O₂-treated BMDMs (Fig. 9h, i). To further evaluate the role of IL-10 on angiogenesis in vivo, the mice were treated with 2A5 or negative control, HRPN (IgG1). ELISA demonstrated that 2A5 blocked the increase of LSI-induced IL-10 release in the endplates at 4 weeks after surgery (Fig. 9j). The sprouting of CD31⁺Emcn⁺ blood vessels in the endplates was significantly inhibited by 2A5 treatment relative to IgG1 treatment, as indicated by immunofluorescent staining (Fig. 9k, l). Predictably, 2A5 administration prevented the sclerosis of endplates in LSI surgery mice, according to micro CT analysis (Fig. 9m, n) and Safranin O and fast green staining (Fig. 9o, p). Furthermore, the presence of Osx⁺ osteoblast progenitors in the endplates was also inhibited by 2A5 treatment in LSI surgery mice, as determined by immunohistochemical staining (Fig. 9q, r). Vitally, the innervation of CGRP⁺ fibers was significantly prevented by 2A5 treatment relative to IgG1 treatment (Fig. 9s, t). Together, these findings reveal that IL-10 is pivotal for senescent-like macrophage-induced angiogenesis and sclerosis in degenerative endplates. The TGF-β has been reported to be involved in endplate sclerosis[37]. We conducted the co-immunofluorescent staining of TGF-β with F4/80 in vivo. The data showed that the TGF-β could co-localize with F4/80 in the endplates of the LSI group or aged group (Supplementary Fig. 18a). Additionally, western blot also showed that the total level of TGF-β in the senescent-like macrophages increased in vitro relative to naive macrophages (Supplementary Fig. 18b, c).

## IL-10 stimulates pSTAT3 signaling to induce angiogenesis

To explore the pathway of IL-10-mediated angiogenesis, we first selected several angiogenesis-associated factors based on literature, including VEGFA, MMP2, PDGFB, MMP9, FGF2, VEC, and HIF-1α[40,41]. qRT-PCR demonstrated that recombinant IL-10 significantly stimulated the mRNA expression of Vegfa (VEGFA), Mmp2 (MMP2), and Pdgfb (PDGFB), but inhibited or slightly increased the mRNA expression of Mmp9 (MMP9), Fgf2 (FGF2), Hif1a (HIF-1α), and Cdh5 (VEC) in endothelial cells (Fig. 10a). According to western blot analysis, the protein levels of VEGFA, MMP2, and PDGFB were elevated significantly by recombinant IL-10 in endothelial cells (Fig. 10b, c). Given the crucial

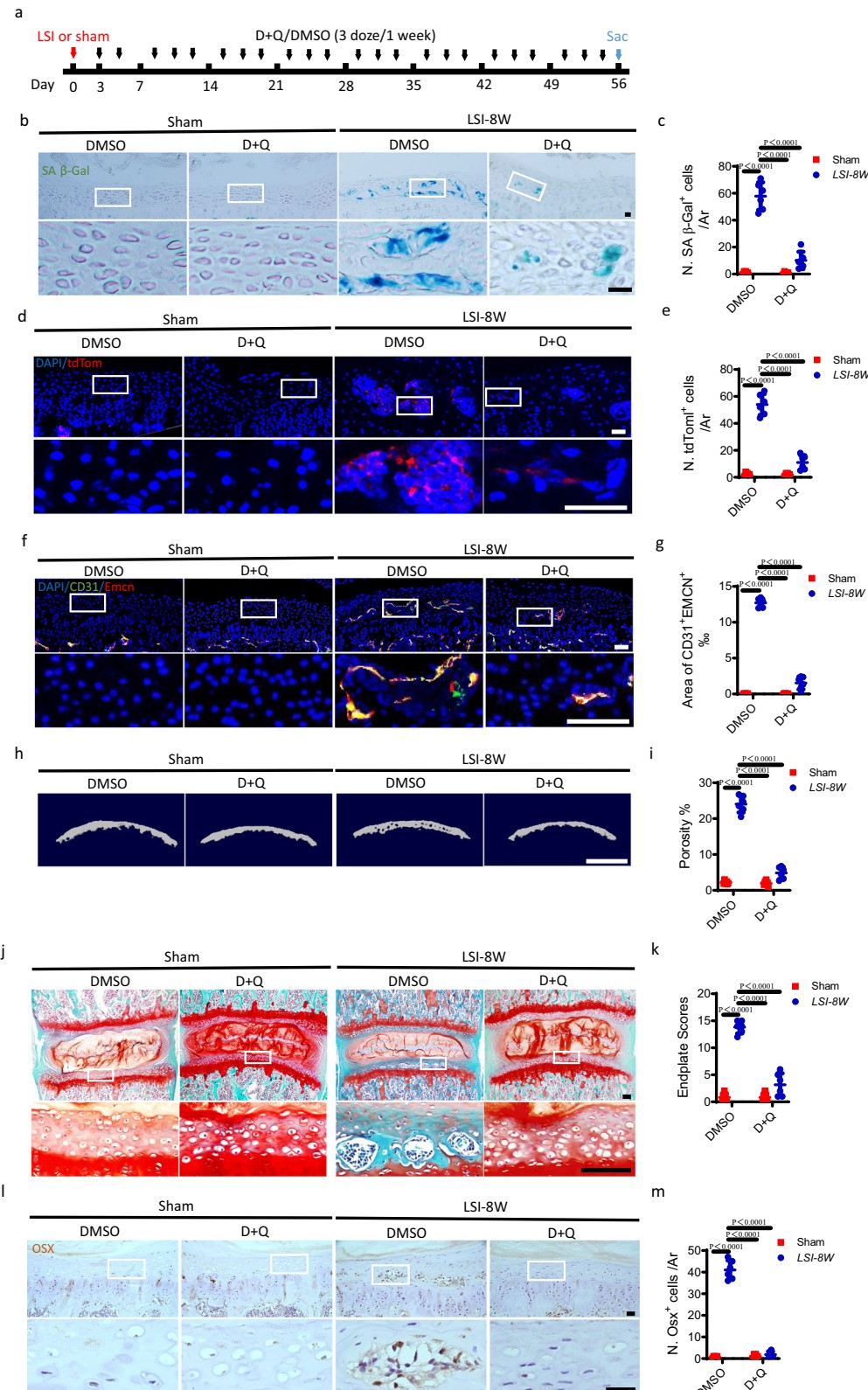

role of the STAT3/pSTAT3 pathway in angiogenesis[42,43], endothelial cells were treated with Stattic, a nonpeptidic selective STAT3 inhibitor. Recombinant IL-10-mediated overexpression of VEGFA, MMP2, and PDGFB were partially abolished by administration of Stattic, as indicated by western blot (Fig. 10b, c). Importantly, recombinant IL-10 significantly promoted tube formation of endothelial cells relative to that of vehicle treatment, while this effect was abrogated by Stattic

administration (Fig. 10d, e). To better determine the mechanism by which IL-10/pSTAT3 regulates the production of VEGFA, MMP2, and PDGFB, a Chromatin Immunoprecipitation (ChIP) assay was performed to detect the probable binding sites of pSTAT3 on the Vegfa (VEGFA), Mmp2 (MMP2), and Pdgfb (PDGFB) promoters. ChIP assay revealed that IL-10 significantly increased the specific binding of pSTAT3 to the Vegfa, Mmp2, and Pdgfb promoters in endothelial cells (Fig. 10f, g).

**Fig. 3 | The sclerosis of endplates is delayed by clearance of senescent cells in LSI mice. a–k** Treatment regimen with D + Q or DMSO (black arrows) in mice after LSI surgery or sham surgery. Red arrow indicates the day of surgery. Blue arrow indicates the day of euthanasia. **b** Representative images of coronal mouse caudal endplate sections of L4/5 stained for SA-βGal (green). Scale bars, 50 μm. **c** Quantitative analysis of **b**. **d** Representative immunofluorescent images of tdTom⁺ cells (red) and DAPI (blue) staining of nuclei at 8 weeks after LSI or sham surgery in p16^tdTom mice. Scale bars, 50 μm. **e** Quantitative analysis of **d**. **f** Representative immunofluorescent images of CD31 (green), Emcn (red) staining, and DAPI (blue) staining of nuclei. Scale bars, 50 μm. **g** Permillage of CD31⁺Emcn⁺

area in the endplates of **f**. **h** Representative three-dimensional μCT images of the caudal endplates of L4/5 (coronal view). Scale bars, 500 μm. **i** Quantitative analysis of the total porosity of **h**. **j** Representative images of safranin O and fast green staining, proteoglycan (red) and cavities (green). Scale bars, 50 μm. **k** Endplate scores as an indication of endplate degeneration based on safranin O and fast green staining. **l** Representative immunohistochemical images of Osterix (Osx). Scale bars, 50 μm. **m** Quantitative analysis of the number of Osx⁺ cells of **l**. $n = 6$ per group. Data are represented as means ± standard deviations, as determined by One-way ANOVA. Source data are provided as a Source Data file.

Together, IL-10 can raise the level of pSTAT3 in endothelial cells, and then pSTAT3 directly binds to the promoters of Vegfa, Mmp2, and Pdgfb to promote their production.

## CD68⁺P16⁺ cells correlate with the sprouting of CD31⁺Emcn⁺ blood vessels in human degenerated endplates

To clarify the involvement of senescent-like macrophages in the angiogenesis of endplates clinically, we evaluated the pathological alterations in the endplate lesions of patients with LBP history (Supplementary Table 1). Severe endplate lesions with sclerosis were observed in degenerative patients with a history of frequent LBP, whereas the cartilaginous integrity was retained in healthier endplate samples (Supplementary Fig. 19a). The increased endplate scores were also found in degenerative endplates relative to healthier endplate samples (Supplementary Fig. 19b). Immunofluorescent staining revealed that the number of CD68⁺P16⁺ cells increased in the porous areas (lesions) of endplates from patients with LBP history relative to healthier endplate samples (Supplementary Fig. 19c, d). Importantly, correlation analysis revealed that the number of CD68⁺P16⁺ cells was significantly positively associated with the visual analog scale (VAS) score of patients (Supplementary Fig. 19e). ELISA assay demonstrated that the level of IL-10 significantly increased in the endplate lesions relative to healthier endplate samples (Supplementary Fig. 19f). Intriguingly, the sprouting of CD31⁺Emcn⁺ blood vessels also elevated in the endplate lesions, as evidenced by immunofluorescent staining (Supplementary Fig. 19g, h). These results imply that senescent-like macrophages are potentially related to angiogenesis and sclerosis in degenerative endplates of human samples.

## Discussion

We have previously demonstrated that the degenerative endplates undergo sclerosis during degeneration with the increased porous area, where sensory nerves sprout. The algogenic substances are then sensed by the innervated sensory nerves to cause spinal pain[11]. Endplate sclerosis is thus a main source of low back pain. However, the underlying mechanism of endplate sclerosis is still unclear. The coupling of angiogenesis and osteogenesis has gained widespread recognition as a crucial component for bone physiology and regeneration, especially since the identification of CD31^hiEmcn^hi capillary subtype with unique morphological, molecular, and functional characteristics in the murine skeletal system. These vessels maintain the perivascular osteoprogenitors to coordinate angiogenesis and osteogenesis[32,44–46]. In this study, using the mouse model of LSI or aging, we also find the robust sprouting of CD31⁺Emcn⁺ blood vessels in the sclerotic endplates. Crucially, the inhibition of angiogenesis could abrogate the endplate sclerosis process.

We then logically sought to investigate the mechanism of angiogenesis in degenerative endplates. Intriguingly, our findings show that senescent cells accumulate in the sclerotic endplates. Cellular senescence refers to a stable state of growth arrest in response to various stresses, which is a complex and multistep biological process. By secreting SASP, senescent cells could aid in the reprogramming of their neighboring cells[47]. It has been reported that pathological angiogenesis engages cellular senescence in diabetic retinopathy[48].

Similarly, our results suggest that the elimination of senescent cells in degenerative endplates suppresses pathological angiogenesis, which then inhibits the endplate sclerotic process. It has been recently reported an important beneficial role for senescent cells in epithelial regeneration[49], showing that the role of cellular senescence is dual-sided. The vascularization is noted in the degenerated discs[50–52]. The expression levels of VEGF-A and CD31 are elevated in degenerative discs of patients, which could trigger angiogenesis[53]. Additionally, the level of angiogenic factor angiopoietin-2 is reported to increase in degenerative disc in patients[54]. Among the nucleus pulposus cells (NPCs), the fibroNPCs showed the highest score of angiogenesis compared to other types of NPC, as evidenced by Single-cell Transcriptome Profiling[55]. Although we did not observe obvious angiogenesis in the nucleus pulposus tissue, the angiogenic factors from NPCs might also be involved in the angiogenesis process of the endplates. Usually, cartilage is a physiologically avascular tissue, but some angiogenic factors could be released from the extracellular matrix and be activated to participate in the angiogenesis process, such as TGF-β1[56]. Although the administration of IL-10 neutralizing antibody demonstrated that IL-10 was crucial to induce angiogenesis in sclerotic endplates, the composition of degenerative cartilage endplates may also be involved in the angiogenesis of endplates.

Different from acute injury-induced senescence, chronic senescence does not have a specific program, but it is a random process. Multiple and persistent stresses that act on tissues and organs may induce chronic senescence[57,58]. In our study, the LSI model does not directly affect the endplates but induces endplate degeneration through unstable mechanical loading. Therefore, the LSI model is preferring to the mechanism of accelerating chronic senescence. In both the LSI model and the naturally aged mouse model, multiple and persistent factors such as aberrant mechanical loading, oxidative stress, cytotoxicity, and mitochondrial dysfunction may be the potential mechanisms of recruitment of senescent-like macrophages. However, the specific mechanisms still need to be further explored in the future. Immune senescence is an important part of cellular senescence, which accelerates aging and tissue damage in systemic organs[59]. According to reports, accelerated immune senescence occurs in patients with chronic kidney disease, leading to chronic inflammation and complications such as cardiovascular disease and infection[60,61]. Senescent-like macrophages accumulate in organisms with age, which may exacerbate radiation-induced pulmonary fibrosis[62], obesity-induced inflammation[63], and skeletal aging[64]. Impressively, senescent-like macrophages are involved in pathological angiogenesis in eyes[65]. Our findings indicate that senescent-like macrophages contribute to endplate sclerosis in both the LSI mouse model and the aging mouse model. The eradication of senescent-like macrophages genetically or pharmaceutically prevents angiogenesis and sclerosis in the endplates during degeneration. Since macrophages are precursors of osteoclasts, when macrophages are deleted by CL injection or knockout in iDTR^ΔLyz2 mice, osteoclasts are inevitably reduced. Thus, the delayed endplate degeneration by the removal of macrophages may partly be attributed to the inhibition of osteoclast activity.

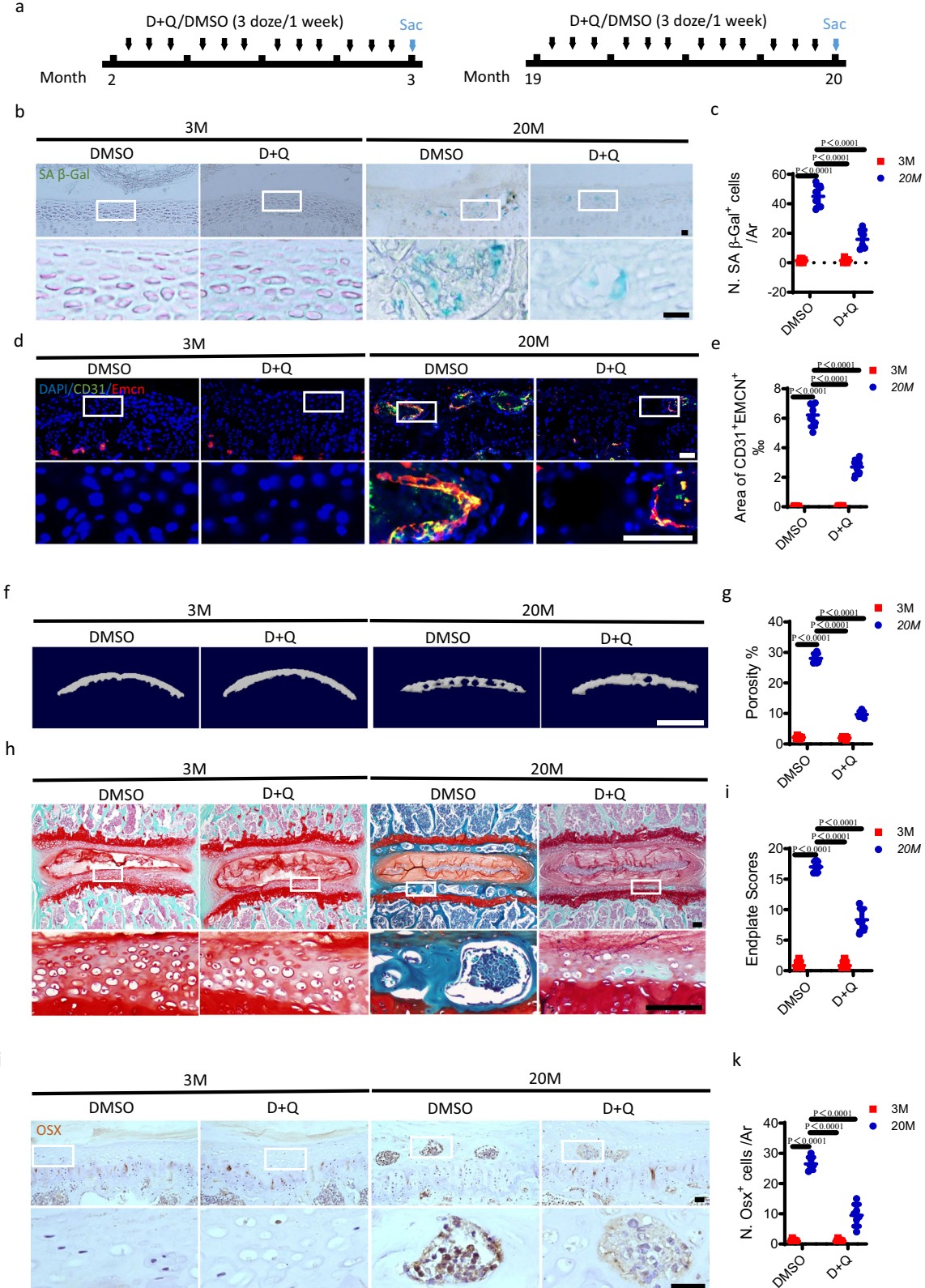

Although SA-β-gal and P16 were commonly used markers for cellular senescence, they could also be detected in non-senescent cells[66-69]. Therefore, due to the lack of a specific marker, to determine whether macrophages undergo senescence in degenerated endplates, we comprehensively evaluate multiple markers, including SA-β-gal, P16, DNA damage markers (SADS, γH2A.X), and SASP[70,71]. Recent research has shown that senescent cells exploit anti-apoptotic

machinery to survive[72-74]. We used senolytic to eliminate senescent cells by inhibiting the anti-apoptotic proteins. We conducted the western blot of the anti-apoptotic Bcl-2 protein and pro-apoptotic protein Bax on the senescent BMDMs. The results demonstrate that the senescence in macrophages is accompanied by an increase in the expression level of the anti-apoptotic Bcl-2 protein. Additionally, mitochondrial dysfunction and cellular senescence are hallmarks of

**Fig. 4 | The sclerosis of endplates is delayed by clearance of senescent cells in aged mice. a**–**k** Treatment regimen with D + Q or DMSO (black arrows) in 3-month-old or 20-month-old mice. Blue arrow indicates the day of euthanasia. **b** Representative images of coronal mouse caudal endplate sections of L4/5 stained for SA-βGal (green). Scale bars, 50 μm. **c** Quantitative analysis of the number of SA-βGal⁺ cells in the endplates of **b**. **d** Representative immunofluorescent images of CD31 (green), Emcn (red) staining and DAPI (blue) staining of nuclei in the endplates. Scale bars, 50 μm. **e** Permillage of CD31⁺Emcn⁺ area in endplates of **d**. **f** Representative three-dimensional μCT images of the mouse caudal endplates of L4/5 (coronal view). Scale bars, 500 μm. **g** Quantitative analysis of the total porosity of **f**. **h** Representative images of safranin O and fast green staining in the endplates, proteoglycan (red) and cavities (green). Scale bars, 50 μm. **i** Endplate scores as an indication of endplate degeneration based on safranin O and fast green staining. **j** Representative immunohistochemical images of Osterix (Osx) in the endplates. Scale bars, 50 μm. **k** Quantitative analysis of the number of Osx⁺ cells in the endplates of **j**. *n* = 6 per group. Data are represented as means ± standard deviations, as determined by One-way ANOVA. Source data are provided as a Source Data file.

aging. Although mitochondrial dysfunction and cellular senescence are different cellular states, they are closely interconnected. Senescence could drive mitochondrial dysfunction, for example, 1) mitochondrial DNA (mtDNA) damage could cause mitochondrial dysfunction, 2) mitochondria of senescent cells undergo structural changes associated with an increase in size and volume, 3) dysregulated nutrient sensing pathways (such as AMPK, SIRT, and mTOR) in senescence could induce mitochondrial dysfunction, 4) activation of IP3P led to $Ca^{2+}$ release to induce mitochondrial $Ca^{2+}$ overload. Meanwhile, mitochondrial dysfunction governs the senescent phenotype, for example, 1) impaired mitochondria produce excessive ROS to cause DNA damage and cell cycle arrest, 2) reduced $NAD^+/NADH$ phosphorylates and/or stabilizes p53 and p16INK4 mRNA, 3) dysfunctional mitochondria contribute to innate immune response activation to produce inflammation, 4) multiple antiapoptotic pathways are upregulated in senescence, presumably in response to partial mitochondrial membrane permeabilization[75]. Mitochondrial dysfunction may also be implicated in the senescence during endplate degeneration.

Considering the active state of senescent cells metabolically and transcriptionally, we evaluate the change of immune response in senescent-like macrophages and find a considerable alteration in various cytokines, specifically the production of IL-10. In particular, the supernatant of senescent-like macrophages significantly promotes tube formation of endothelial cells, while the neutralizing antibody to IL-10 inhibits angiogenesis in vitro and in vivo. Predictably, the IL-10 neutralization also delays the osteogenesis and sclerosis of endplates in the LSI model. Although IL-10 has been reported to decrease osteoprotein synthesis of murine bone marrow cells[76], it has also been shown to promote osteogenic differentiation of MC3T3 or dental pulp stem cells[77,78]. IL-10 may therefore exert a dual regulatory effect on endplate sclerosis. Our findings imply that the senescent-like macrophages are the main source of IL-10. However, IL-10 may arise from cells other than senescent-like macrophages, like bone marrow cells[79]. In addition, IL-10 may not be the only factor responsible for the angiogenic and sclerotic condition in endplate degeneration. As indicated by our data, CCL2 expression was also elevated in degenerative endplates and senescent-like macrophages, which may have the ability to stimulate angiogenesis via activating Ets-1[80] and VEGF[81]. According to a recent report, the dysregulation of ATP binding cassette transporter ABCA1 in senescent-like macrophages may contribute to pathological angiogenesis[82]. Additionally, the TGF-β has been reported to be involved in endplate sclerosis[37]. Our data indicates the increase of total TGF-β in the senescent-like macrophages. However, the latent TGF-β was mainly stored in the extracellular matrix[83]. It has been reported that the level of active TGF-β increased in the endplates persistently post-LSI surgery, but the total TGF-β only increased in the endplates at 2 weeks post-LSI surgery[37]. Together, these suggest that the TGF-β released by senescent macrophage might participate in the endplate sclerosis, but may not be the most important source of active TGF-β.

An estimate of macrophage activation at a certain point in space and time is referred to as macrophage polarization. As a review states, the polarized state of macrophage is dynamic across time and there is

no evidence to support dualistic models of macrophage polarization. Due to the lack of clearly defined criteria for scoring phenotypes, the use of labels M1 and M2 is still somewhat debatable. So, macrophage polarization appears complex, for instance, macrophage polarization is occasionally more M2-like but with M1-associated gene expression in tumors[84–87]. Despite its shortcomings, there is no denying that macrophage polarization remains a crucial categorization for research on macrophages. Our results reveal that the majority of the P16-positive cells expressed the M2 polarization marker CD206, as opposed to the M1 polarization marker iNOS in the endplates of LSI surgery mice and aged mice. This suggests that most senescent-like macrophages tend to exhibit the M2 marker. Similarly, recent research has shown that senescent-like macrophages display high levels of M2 markers[65,88]. It has also been reported that senescent-like macrophages differ from M1 and M2 despite expressing markers of macrophage polarization[63]. Co-immunofluorescence of IL-10 with CD206 or iNOS showed that the cells expressing IL-10 largely co-localized with CD206⁺ cells rather than iNOS⁺ cells. Although IL-10 is considered to be produced mainly by M2-type macrophages, it has also been reported that senescent-like macrophages secrete high levels of IL-10[25]. A vital finding was that the number of IL-10⁺F4/80⁺ or IL-10⁺P16⁺ in the endplates of the LSI group was significantly reduced in the Cdkn2a$^{ΔLyz2}$ mice. Together, the macrophage is senescent in the degenerated endplates and tends to exhibit more M2-like marker. The senescent state of macrophages does not contradict the increased IL-10 release.

Previously, we reported that osteoclasts can boost the growth of CGRP⁺ nociceptive sensory nerves into the sclerotic endplates by secreting netrin-1. Then, sensory innervation is involved in the mediation and onset of spinal pain[11]. Interestingly, the specific knockout of Cdkn2a in macrophages in this study inhibits the expansion of CGRP⁺ sensory nerves in the endplates, when abrogating the process of endplate sclerosis. Similar to this, the density of CGRP⁺ nerve fibers in the endplates is reduced by the systematic administration of neutralizing antibody to IL-10. According to these findings, the restriction of the senescent-like macrophage activity may potentially prevent endplate sclerosis and sensory innervation, thereby improving both structural disorders and pain perception in the degenerative endplates.

Our previous study[11] and the present work illustrate the presence of angiogenesis, osteoclast activity, and osteogenesis during the degeneration of the endplates. In the present study, we focused on the role of senescent-like macrophage-mediated angiogenesis in the osteogenesis and sclerosis process. Interestingly, the LSI-induced increase in angiogenesis and the number of Trap⁺ osteoclasts are reduced in Cdkn2a$^{ΔLyz2}$ mice. Considering that the endplate is composed of cartilaginous components from the perspective of development, the action of osteoclasts may also require the invasion of blood vessels. The term "angiogenesis osteogenesis coupling" has demonstrated the strong relationship between osteogenesis and angiogenesis. This is confirmed by the data that blocking angiogenesis can hinder the degeneration of the endplates. Together, we believe that the coupled actions of angiogenesis, osteoclast activity, and osteoblast activity lead to endplate sclerosis. When the angiogenesis is inhibited, the osteoclastic and osteogenic activities are also hindered.

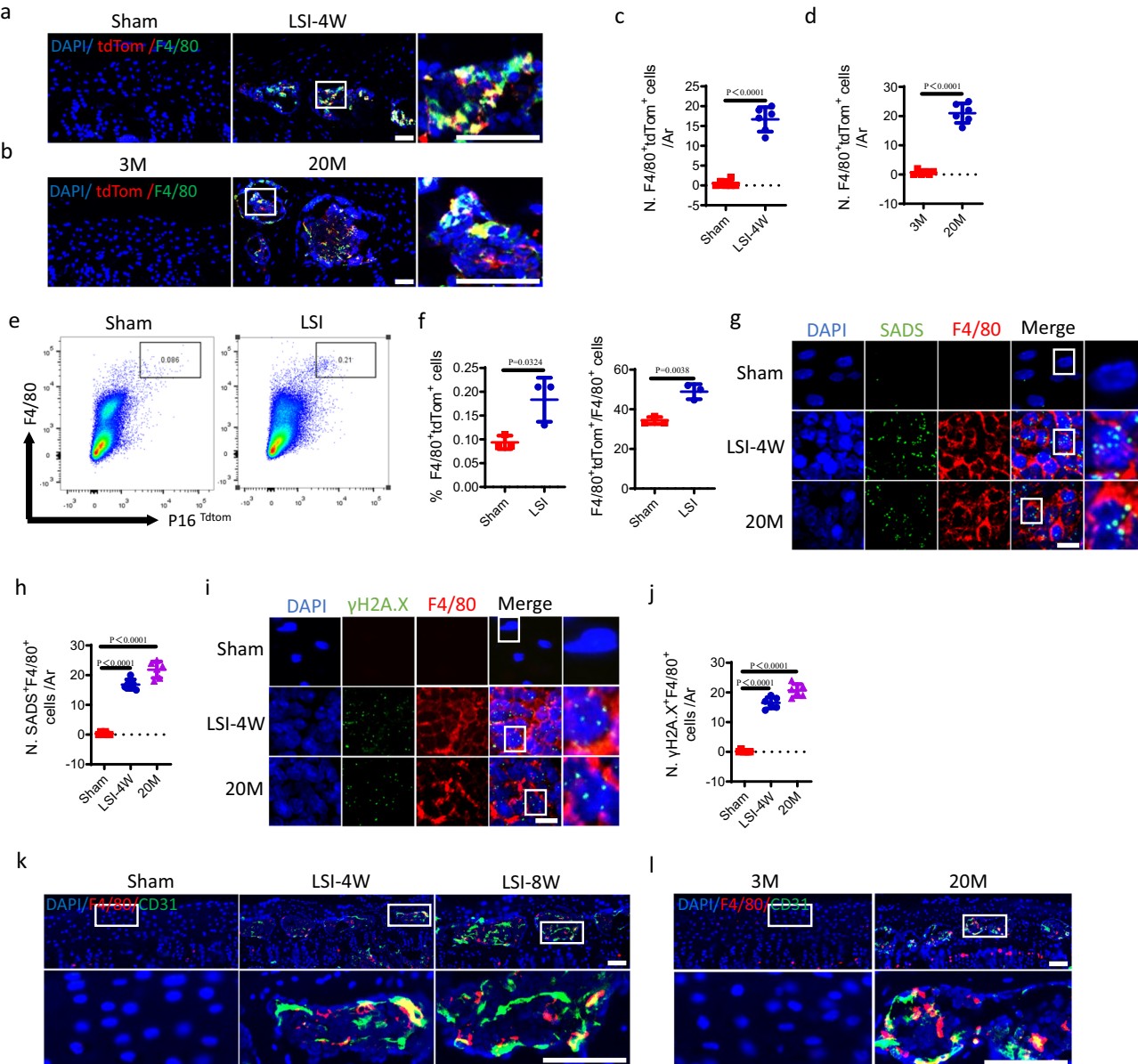

**Fig. 5 | Abundant macrophages undergo senescence in the endplates of LSI mice and aged mice.** Representative immunofluorescent images of tdTom⁺ (red), F4/80⁺ (green) cells and DAPI (blue) staining of nuclei in LSI surgery model (**a**) or aging model (**b**) of cdkn2a (p16)^tdTom mice. Scale bars, 50 µm. **c**, **d** Quantitative analysis of the number of F4/80⁺tdTom⁺ cells in the endplates of **a** and **b**. n = 6 per group. **e** Representative images of flow cytometry analysis. cdkn2a (p16)^tdTom mice underwent sham or LSI surgery. Cells were isolated from the endplates. **f** Quantitative analysis of **e**. n = 3 per group. **g** Representative images of senescence-associated distension of satellites (SADS, green), F4/80 (red) and DAPI (blue) staining of nuclei in the endplates of LSI surgery mice or aged mice by performing simultaneous immunofluorescent staining and immune-fluorescent in situ

hybridization (FISH). Scale bars, 10 µm. **h** Quantitative analysis of the number of F4/80⁺SADS⁺ cells (≥4 SADS/cell) in endplates of **g**. n = 6 per group. **i** Representative immunofluorescent images of γH2A.X (green), F4/80 (red) and DAPI (blue) staining of nuclei in the endplates of LSI surgery mice or aged mice. Scale bars, 10 µm. **j** Quantitative analysis of the number of F4/80⁺γH2A.X⁺ cells in the endplates of **i**. n = 6 per group. Representative immunofluorescent images of F4/80 (red), CD31 (green) and DAPI (blue) staining of nuclei in the endplates at 4, and 8 weeks after LSI or sham surgery (**k**) or in 3-month-old or 20-month-old mice (**l**). Scale bars, 50 µm. n = 3–6 per group. Data are represented as means ± standard deviations, as determined by two-tailed Student's *t* test or One-way ANOVA. Source data are provided as a Source Data file.

## Methods

### Study approval
The use of human tissue samples was approved by the Ethics committee of Zhengzhou University. Patients gave written informed consent for the use of the samples. All animal experiments were approved by the Animal Care and Use Committee of Zhengzhou University Animal Facility and in compliance with the relevant laws. All animal experiments complied with the ARRIVE guidelines for reporting animal experiments.

### Human subjects
Human endplate samples were obtained from patients undergoing spinal surgery in the Department of Orthopaedics at 1st Affiliated Hospital of Zhengzhou University (Zhengzhou, China). Detailed information about the patients and groups is provided in Supplementary Table 1.

### Animals and treatment
We purchased C57BL/6J (WT) male mice from Charles River Laboratories (China). The L3–L5 spinous processes, the

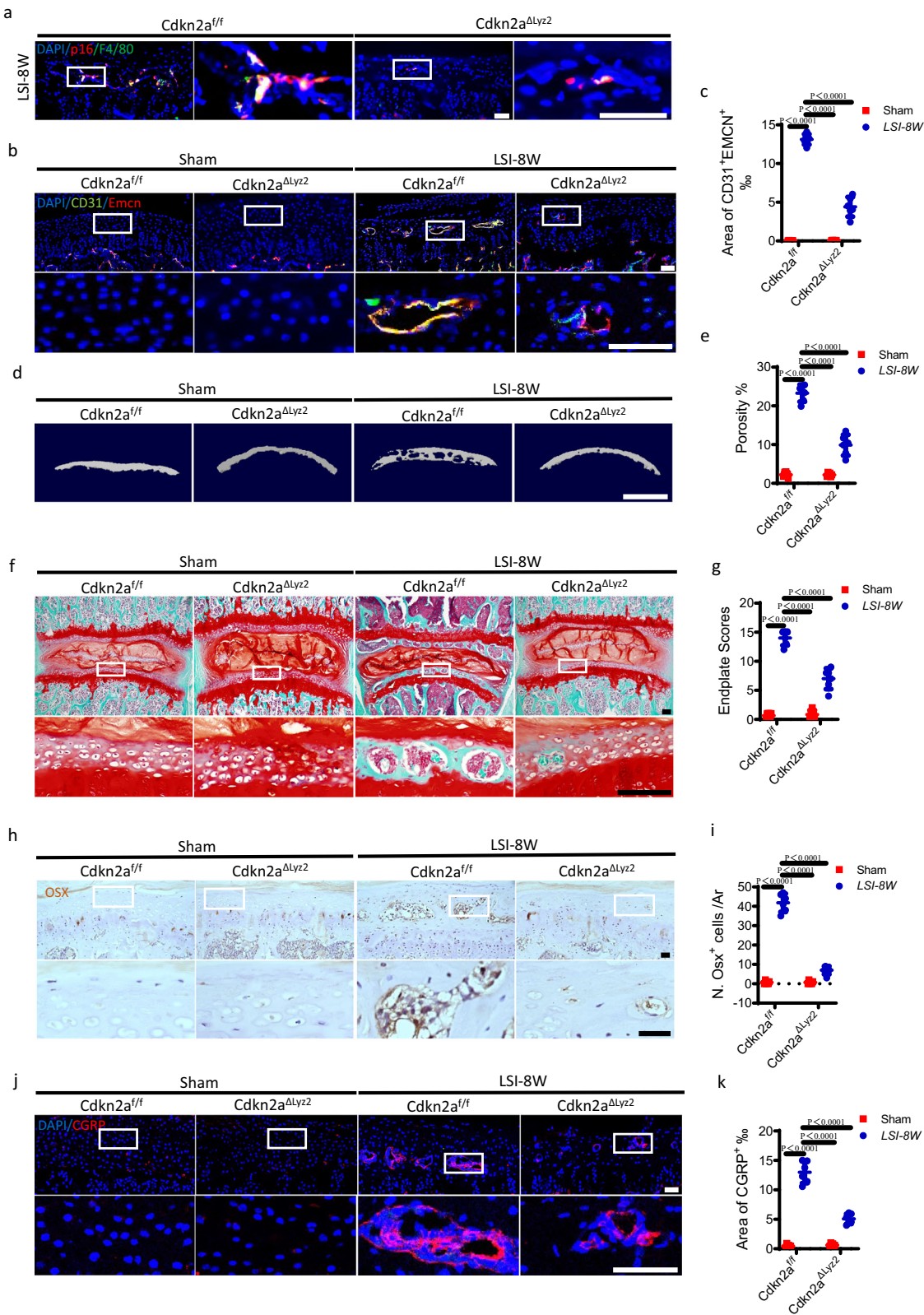

supraspinous ligaments, and the interspinous ligaments were removed to create the Lumbar Spine Instability (LSI) model after undergoing general anesthesia. Only the posterior paravertebral muscles of L3-L5 vertebrae were detached in the sham group mice[11]. For the time-course experiments, the mice were euthanized by an overdose of isoflurane at 4 or 8 weeks after LSI or sham surgery. 2- to 19-month-old C57BL/6J (WT) male mice were purchased from

Jiangsu Aniphe Biolaboratory Inc for the aging-induced endplate degeneration paradigm. Then, the mice were euthanized at 3- or 20-month-old after receiving drugs or vehicles and some mice were euthanized at appropriate age (3, 6, 12, 16-month-old). All mice were maintained at the animal facility of Zhengzhou University Animal Experimental Center under 12-hour light/dark cycle at 20–24 °C and 45–65% humidity. Animals were housed with a maximum of 5 mice

**Fig. 6 | Knockout of cdkn2a (p16) in Lyz2⁺ cells abrogates LSI-induced endplate sclerosis. a** Representative immunofluorescent images of p16⁺ (red), F4/80⁺ (green) cells and DAPI (blue) staining of nuclei in mouse caudal endplates of L4/5 in Cdkn2a^ΔLyz2 or Cdkn2a^f/f mice at 8 weeks after LSI or sham surgery. Scale bars, 50 μm. **b** Representative immunofluorescent images of CD31 (green), Emcn (red) and DAPI (blue) staining of nuclei in the endplates of Cdkn2a^ΔLyz2 or Cdkn2a^f/f mice after LSI or sham surgery. Scale bars, 50 μm. **c** Permillage of CD31⁺Emcn⁺ area in the endplates of **b**. **d** Representative μCT images of the caudal endplates of L4/5 (coronal view) in Cdkn2a^ΔLyz2 or Cdkn2a^f/f mice at 8 weeks after LSI or sham surgery. Scale bars, 500 μm. **e** Quantitative analysis of the total porosity of **d**. **f** Representative images of safranin O and fast green staining of endplates in Cdkn2a^ΔLyz2 or Cdkn2a^f/f mice at 8 weeks after LSI or sham surgery. Scale bars, 50 μm. **g** Endplate scores of the endplates of **f**. **h** Representative immunohistochemical images of Osterix (Osx) in the endplates of Cdkn2a^ΔLyz2 or Cdkn2a^f/f mice at 8 weeks after LSI or sham surgery. Scale bars, 50 μm. **i** Quantitative analysis of the number of Osx⁺ cells in the endplates of **h**. **j** Representative images of immunofluorescent analysis of CGRP⁺ sensory nerves (red) and DAPI (blue) staining of nuclei in the endplates of Cdkn2a^ΔLyz2 or Cdkn2a^f/f mice after LSI or sham surgery. Scale bars, 50 μm. **k** Permillage of CGRP⁺ area in the endplates of **l**. *n* = 6 per group. Data are represented as means ± standard deviations, as determined by One-way ANOVA. Source data are provided as a Source Data file.

per cage Female mice were only used for breeding. Only male mice were used for further experiments (i.e. histology, etc.).

The cdkn2a-(luc-tdtomato-CreERT2) mouse strain (p16^tdTom) (NM-CKO-200015), Lyz2-Cre mouse strain (NMX-KI-192007), and cyclin-dependent kinase inhibitor 2A (Cdkn2a) floxed mouse strain (NM-CKO-200015) were purchased from Shanghai Model Organisms. The B6-iDTR mouse strain (JAX, #007900) was purchased from the Jackson Laboratory. To trace p16⁺ cells, Tamoxifen (75 mg/kg, i.p.) was administered to p16^tdTom mice at designed time points. Heterozygous Lyz2-Cre mice were crossed with Cdkn2a^flox/flox mice; the posterity was intercrossed to produce the following genotypes: wild-type (WT), Lyz2-Cre (expressing Cre recombinase under the control of the Lyz2 promoter), Cdkn2a^flox/flox (homozygous for the Cdkn2a flox allele, denoted as Cdkn2a^f/f) and Lyz2-Cre; Cdkn2a^flox/flox (exhibiting conditional knockout of Cdkn2a in Lyz2⁺ cells, denoted as Cdkn2a^ΔLyz2). Heterozygous Lyz2-Cre mice were interbred with B6-iDTR mice; the posterity was intercrossed to obtain the following genotypes: WT, Lyz2-Cre, B6-iDTR (containing the iDTR allele, designated as iDTR), Lyz2-Cre; B6-iDTR (referred to as iDTR^ΔLyz2) mice. To delete Lyz2⁺ cells, diphtheria toxin (DT, 25 μg/kg, i.p.) was administrated to iDTR^ΔLyz2 mice.

The genotypes of the mice were determined by PCR analyses of genomic DNA, which was extracted from mouse tails within the following primers: p16^tdTom, P1: TGTGTGTAAGAAGAATTCCAAGGC, P2: GAACGCAAATATCGCACGATG, P3: ATAGGGCTTCTTTCTTGGGTCC, P4: TCACGTTCATTATAAATGTCGTTCG; Lyz2-Cre, P1: CCCAGAAAT GCCAGATTACG, P2: CTTGGGCTGCCAGAATTTCTC, P3: TTACAGT CGGCCAGGCTGAC; B6-iDTR, P1: GCGAAGAGT TTGTCCTCA ACC, P2: AAAGTCGCTCTGAGTTGTTAT, P3: GGAGCGGGAGAAATGGATATG; Cdkn2a loxP allele: P1: AGGGAGGGAACATTACTATTTT, P2: GAACG TTGCCCATCATCAT. For the senolytic administration, mice received a cocktail containing 5 mg/kg Dasatinib (A3017, Apexbio) and 50 mg/kg Quercetin (N1841, Apexbio) or the equivalent volume of Dimethyl sulfoxide via oral gavage 3 times weekly until euthanized. For macrophage depletion experiments, 200 μl of clodronate liposomes or empty liposomes (5 mg/ml, from Vrije University, Netherlands, YEA-SEN, Shanghai, China) were injected intraperitoneally at the beginning, and 100 μl of clodronate liposomes or empty liposomes twice weekly after that until euthanized. To neutralize IL-10 in vivo, mice were randomized into cytokine-blocking mAb-treatment (IL-10) or HRPN rat-IgG1 groups. 500 μg of HRPN (rIgG1) or IL-10-blocking mAb (JES5-2A5, BioXcell) was administrated intraperitoneally twice weekly for 2 weeks, then 200 μg per dose for 2 more weeks. For the treatment of SU5416, a pan-VEGF receptor tyrosine kinase inhibitor, 2 mg/kg of SU5416 or DMSO was injected intraperitoneally twice a week immediately after LSI surgery or sham surgery until euthanized.

## Micro-CT analysis

Mice were euthanized by an overdose of isoflurane and then collected spines were fixed in 10% buffered formalin for 48 hours, and the Micro-CT (μCT) scanning (Bruker Skyscan1172) was performed on the spines with parameters of 55 kV (voltage) and 181 μA (current) at 13.0 μm resolution. Images were reconstructed using the NRecon v1.6 and analysis was performed using CTAn v1.9 (Skyscan US, San Jose, CA). We selected a region of interest (ROI) by contouring the boundary of the L4-L5 vertebral unit (coronal view, caudal endplates). Total porosity for the caudal endplates was assessed in the 3-dimensional datasets. Six consecutive coronal-oriented images were used to illustrate the 3-dimensional model of the caudal endplates using CTVol v2.0 (Skyscan US).

## Histochemistry, immunohistochemistry, and histomorphometry

After fixation for 48 h, the spine samples were decalcified in 0.5 M EDTA (pH 8.0@25 °C, Servicebio) for 2 weeks. The tissues were embedded in paraffin or optimal cutting temperature (OCT). 4μm-thick coronal-oriented sections of the caudal endplate of L4/5 were processed for Safranin O and fast green, TRAP (Servicebio), and immunohistochemical staining. To evaluate endplate scores, a standardized scoring system was introduced as described previously[89,90]. In detail, structural integrity of lumbar spinal endplates was scored based on the presence of number of cells (chondrocyte clusters), structural disorganization, clefts, microfracture and neovascularization, with 0 indicating intact morphology and higher score indicating more severe damage. Degeneration was indicated by the presence of new bone formation, bony sclerosis, physiologic vessels, scar formation and tissue defects, with 0 indicating absence and 1 indicating presence. Evaluation was performed in a blinded manner by trained assessors to minimize bias. Overall endplate scores were determined by summing individual scores across all parameters, providing a comprehensive assessment of endplate health and morphology. 10-μm-thick coronal-oriented sections of the caudal endplate of L4/5 were used for immunofluorescent staining, and SA-βGal staining using a standard protocol. The sections were incubated with primary antibodies against mouse Endomucin (1:50, sc-65495, Santa Cruz Biotechnology; 1:50, ab106100, Abcam), CD31 (1:100, AF3628, R&D; 1:200, ab76533, Abcam), F4/80 (1:100, 14-4801-82, eBioscience), γH2A.X (1:100, ab81299, Abcam), p16 (1:100, AF5484, Affinity; 1:400, sc-377412, Santa Cruz Biotechnology), CD68 (1:50, ab955, Abcam; 1:50, 28058-1-AP, Proteintech), IL-10 (1:50, sc-365858, Santa Cruz Biotechnology, Inc.), CD206 (1:50, sc-58986, Santa Cruz Biotechnology, Inc.), iNOS (1:50, sc-7271, Santa Cruz Biotechnology, Inc.), TGFβ (1:50, sc-130348, Santa Cruz Biotechnology, Inc.), RFP (1:250, 5F8, Chromotek GmbH, Planegg-Martinsried, Germany), HMGB1 (1:300, ab18256, Abcam), CGRP (1:100, ab81887, Abcam), osterix (1:200, ab22552, Abcam), and osteocalcin (1:200, M188, Takara) overnight at 4 °C. Then, the sections were incubated with secondary antibodies for 1 h at room temperature (RT). Nuclei were labeled with 4′6-Diamidino-2-phenylindole-dihydrochloride (DAPI, Servicebio, G1012) for immunofluorescence assay. For immunohistochemistry, a streptavidin horseradish peroxidase detection kit (Vector) was subsequently used to detect the immunoactivity, followed by counterstaining with hematoxylin (Servicebio). The senescence βGal staining kit was used according to the manufacturer's instructions (Cell Signaling Technology, Danvers, MA) to identify senescent cells. Sample images were captured with a fluorescent microscope (Olympus, Japan) or confocal microscope (Zeiss

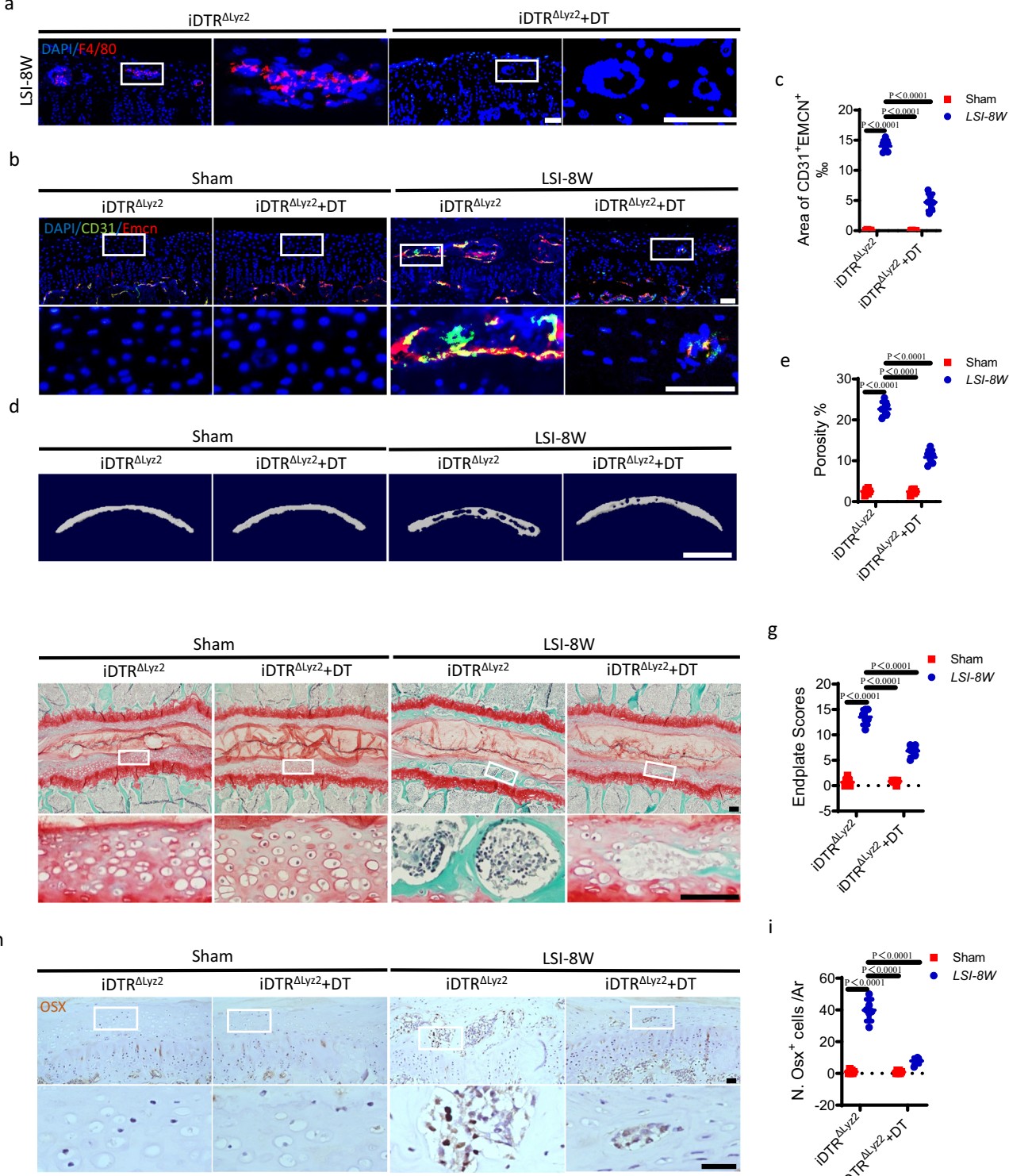

**Fig. 7 | Deletion of macrophages inhibits LSI-induced endplate sclerosis.**
**a** Representative immunofluorescent images of F4/80[+] (red) cells and DAPI (blue) staining of nuclei in mouse caudal endplates of L4/5 in iDTR[ΔLyz2] mice with diphtheria toxin (DT) or vehicle injection at 8 weeks after LSI surgery. Scale bars, 50 μm. **b–i** All the experiments were conducted at 8 weeks after LSI or sham surgery in iDTR[ΔLyz2] mice with DT or vehicle injection. **b** Representative images of immunofluorescent analysis of Emcn[+] (red), CD31[+] (green) cells and DAPI (blue) staining of nuclei in the endplates. Scale bars, 50 μm. **c** Permillage of CD31[+]Emcn[+] area in the

endplates of **b**. **d** Representative μCT images of the caudal endplates of L4/5 (coronal view). Scale bars, 500 μm. **e** Quantitative analysis of the total porosity of **d**. **f** Representative images of safranin O and fast green staining of the endplates. Scale bars, 50 μm. **g** Endplate scores of **f**. **h** Representative immunohistochemical images of Osterix (Osx) in the endplates. Scale bars, 50 μm. **i** Quantitative analysis of the number of Osx[+] cells in the endplates of **h**. $n = 6$ per group. Data are represented as means ± standard deviations, as determined by One-way ANOVA. Source data are provided as a Source Data file.

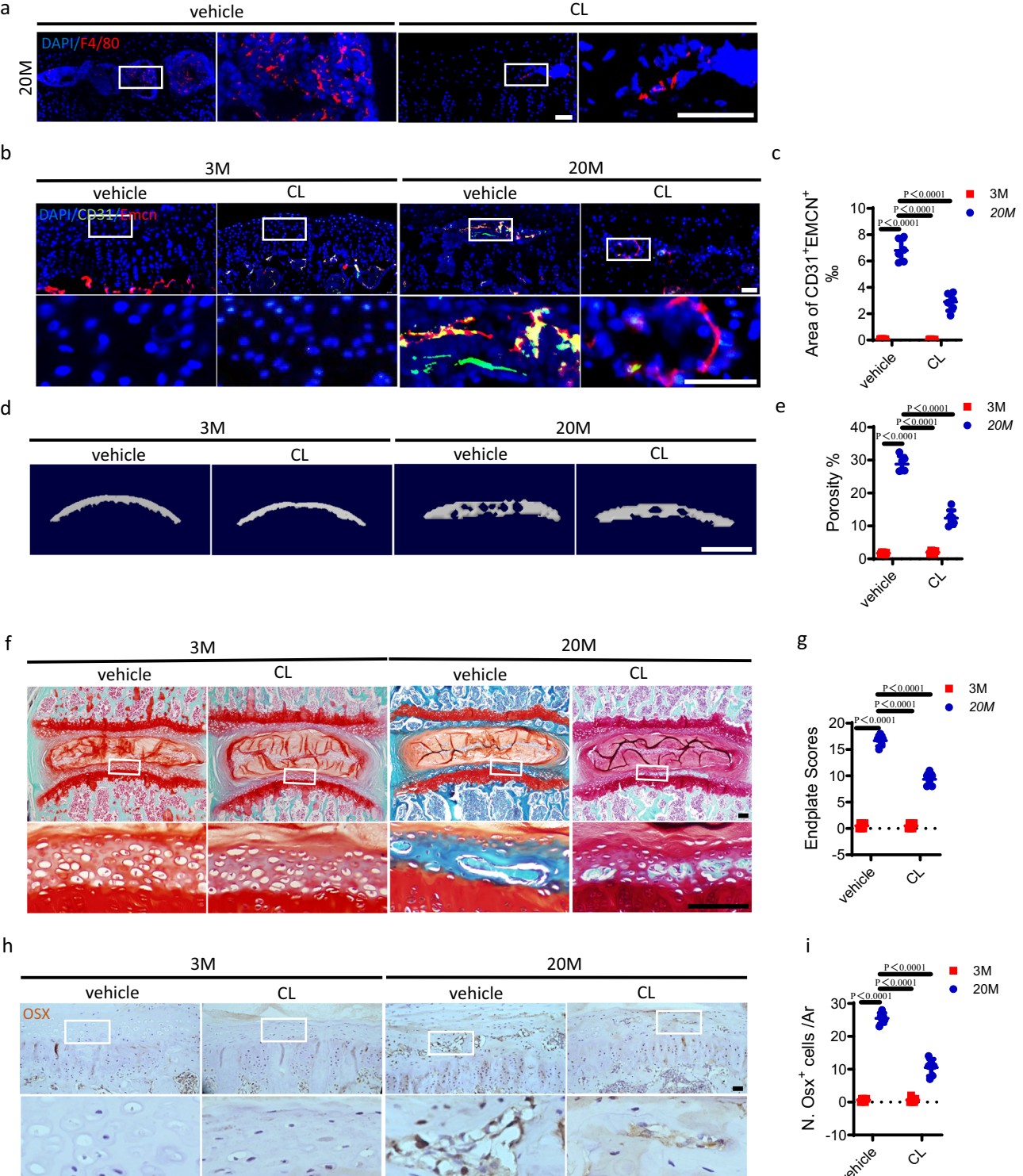

**Fig. 8 | Deletion of macrophages inhibits aging-induced endplate sclerosis.**
**a** Representative immunofluorescent images of F4/80 (red) staining and DAPI (blue) staining of nuclei in the caudal endplates of L4/5 in 20-month-old mice with clodronate liposomes (CL) or vehicle injection. Scale bars, 50 μm. **b–i** All the experiments were conducted in 20-month-old mice or 3-month-old mice with CL or vehicle injection. **b** Representative images of immunofluorescent analysis of CD31 (green), Emcn (red) staining and DAPI (blue) staining of nuclei in the endplates. Scale bars, 50 μm. **c** Permillage of CD31⁺Emcn⁺ area in the endplates of **b**.

**d** Representative μCT images of the caudal endplates of L4/5 (coronal view). Scale bars, 500 μm. **e** Quantitative analysis of the total porosity of **d**. **f** Representative images of safranin O and fast green staining of the endplates. Scale bars, 50 μm. **g** Endplate scores of **f**. **h** Representative images of immunohistochemical staining of Osterix (Osx) in the endplates. Scale bars, 50 μm. **i** Quantitative analysis of the number of Osx⁺ cells in endplates of **h**. $n = 6$ per group. Data are represented as means ± standard deviations, as determined by One-way ANOVA. Source data are provided as a Source Data file.

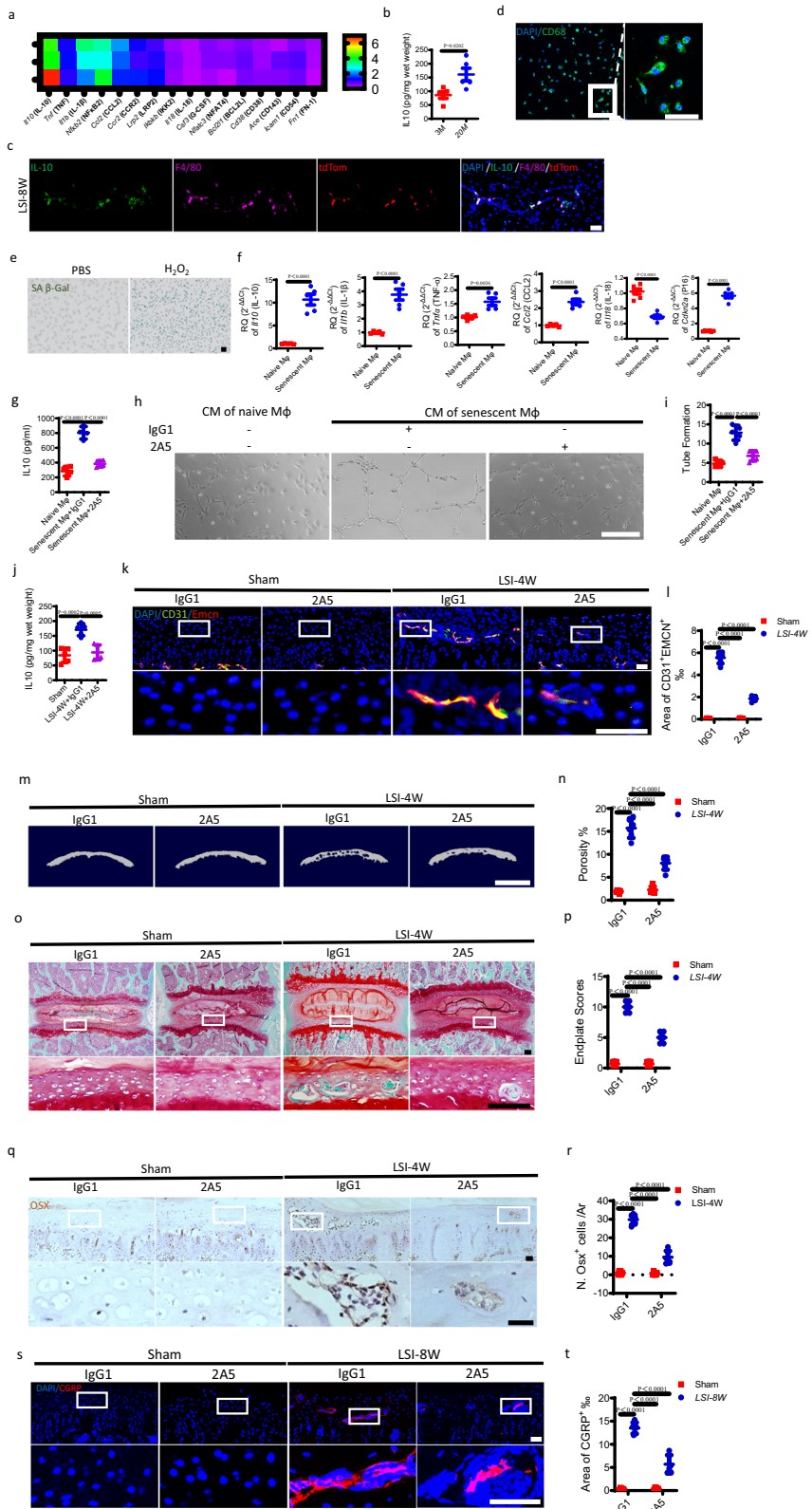

LSM 980). The area is obtained by measuring the target's lower end-plate using Image J software. Quantitative analysis was conducted with ImageJ software (NIH).

**Cell sorting and flow cytometry analysis of cartilage endplate**

After carefully removing the muscle tissue and ligament around the spine, separate and cut the endplates with a surgical blade under a stereomicroscope. Flush the endplates with fluorescence-activated cell sorting (FACS) buffer (5% FBS in PBS), and cut the tissue into 1 mm³ fragments with sterile scissors. The endplates were further digested with 0.25% trypsin (Solarbio) for 20 min and 0.25% Collagenase II (Solarbio) solution for 1.5 h at 37 °C and passed through a 40 μm cell strainer to yield single-cell suspensions. The cells were collected and resuspended in red blood cell lysis buffer (Solarbio) for the lysis of red

**Fig. 9 | IL-10 secreted by senescent-like macrophages contributes to angiogenesis in the degenerative endplates. a** Target mRNA expression levels of lumbar endplates from 20M mice or 3 M mice. Heatmap of top 10 dysregulated cytokines. **b** ELISA analysis of IL-10 concentration in the lysate of lumbar endplates from 20 M mice or 3 M mice. $n = 5$ per group. **c** Representative immunofluorescent images of tdTom (red), IL-10 (green), F4/80 (magenta) staining and DAPI (blue) staining of nuclei in the endplates at 8 weeks after LSI surgery in p16[tdTom] mice. Scale bars, 50 μm. **d–f** The primary BMDMs were harvested from WT mice and treated with 100 μM $H_2O_2$ or PBS for 4 h. **d** Representative images of immunofluorescent analysis of CD68 (green) staining and DAPI (blue) staining of nuclei in the BMDMs. Scale bars, 50 μm. **e** Representative images of BMDMs stained for SA-βGal (green). Scale bars, 50 μm. **f** The mRNA expression levels of Il10 (IL-10), Il1b (IL-1β), Tnfα (TNF-α), Ccl2 (CCL2), Il18 (IL-18), and Cdkn2a (P16) in BMDMs were determined using qRT-PCR. Each group consisted of $n = 5$ for Il10, Il1b, Tnfα, and Ccl2, while Il18 and Cdkn2a had $n = 6$ per group. **g–i** BMDMs were treated with $H_2O_2$ or PBS, plus IL-10-blocking mAb (2A5) or negative control (IgG1) administration. **g** ELISA analysis of IL-10 concentration in the lysate of BMDMs. $n = 4$ per group. **h** Representative tube formation images of endothelial cells treated with conditioned medium (CM) from BMDMS. Scale bars, 50 μm. **i** Quantitative analysis of tube formation of **h**. $n = 5$ per group. **j–t** LSI surgery mice or sham surgery mice were administrated with 2A5 or IgG1. **j** ELISA analysis of IL-10 concentration in the lysate of lumbar endplates. $n = 5$ per group. **k** Representative immunofluorescent images of CD31 (green), Emcn (red) staining and DAPI (blue) staining of nuclei in the endplates. Scale bars, 50 μm. **l** Permillage of CD31⁺Emcn⁺ area in endplates of **k**. $n = 6$ per group. **m** Representative μCT images of the caudal endplates of L4/5 (coronal view). Scale bars, 500 μm. **n** Quantitative analysis of the total porosity of **m**. $n = 6$ per group. **o** Representative images of safranin O and fast green staining of the endplates. Scale bars, 50 μm. **p** Endplate scores of **o**. $n = 6$ per group. **q** Representative immunohistochemical images of Osterix (Osx) in the endplates. Scale bars, 50 μm. **r** Quantitative analysis of the number of Osx⁺ cells in the endplates of **q**. $n = 6$ per group. **s** Representative immunofluorescent images of CGRP⁺ sensory nerves (red) and DAPI (blue) staining of nuclei in the endplates. Scale bars, 50 μm. **t** Permillage of CGRP⁺ area in the endplates of **s**. $n = 6$ per group. Data are represented as means ± standard deviations, as determined by two-tailed Student's $t$ test or One-way ANOVA. Source data are provided as a Source Data file.

blood cells. After washing, cells were resuspended in 100 μl FACS buffer and incubated with Brilliant Violet 421™ anti-mouse F4/80 antibody (1:100, 123131, Biolegend, Inc., San Diego, CA) for 30 min at 4 °C. After washing, cells were analyzed using a BD LSRFortessa flow cytometer. The data were analyzed with FlowJo software (version 10, BD Bioscience).

### Senescence-associated distension of satellites (SADS) analysis

SADS or large-scale peri-centromeric satellite heterochromatin DNA unwinding was recently recognized as a feature of senescent cells in osteocytes, fibroblasts, hepatocytes, glial cells, and numerous other cell types by fluorescence in situ hybridization (FISH)[91–93]. The SADS analysis was conducted as previously described. The spine sections were cross-linked by 4% paraformaldehyde (PFA) for 20 min, then washed in PBS and dehydrated in ethanol (70%, 90%, 100%). After being air dried and denaturized, the sections were hybridized with Cy3-labeled CENPB-specific (ATTCGTTGGAAACGGGA) peptide nucleic acid (PNA) probe (F3002, Panagene Inc., Korea) in a dark room for 2 h at RT. Subsequently, primary antibodies (anti-F4/80, 1:100; 14-4801-82, eBioscience) were incubated at 4 °C overnight. The corresponding secondary antibodies (1:500; Abcam) were incubated for 1 h at RT. The spine sections were then washed and mounted with vectashield DAPI-containing mounting media. The number of decondensed peri-centromeric satellites per cell were assessed by quantification of SADS. The confocal microscope (Zeiss LSM 980) was utilized to visualize SADS (i.e., decondensed/elongated centromeres), and sections were imaged using in-depth Z stacking (a minimum of 40 optical slices with 63× objective) followed by Huygens (SVI) deconvolution. A senescent cell was determined by a cut-off of ≥4 SADS per cell, as previously described[91–93]. Herein, the positive cell was defined as containing ≥4 SADS in nucleus per F4/80⁺ cell. At least 30 nuclei were analyzed for each sample. The number of SADS in each cell was quantified in a blinded fashion.

### Cell culture and treatment

For the isolation of bone marrow-derived macrophages (BMDMs), the femur and tibia of 8-week-old male C57BL/6J mice were collected and both ends of the long bones were cut. BMDMs were flushed from hindlimbs with DMEM (Gibco, Thermo Fisher Scientific) containing 10% FBS (Gibco, Thermo Fisher Scientific). The cell suspension was seeded at $1 \times 10^7$/mL and cultured with DMEM (Gibco, Thermo Fisher Scientific) containing 10% FBS (Gibco, Thermo Fisher Scientific) plus 1% penicillin-streptomycin (Servicebio) overnight. Then, the non-adherent cells were collected and incubated in the medium supplemented with 30 ng/mL M-CSF (Peptrotech) to 80% confluence. Immunofluorescent staining was conducted to identify BMDMs using an antibody to CD68 (ab283654, Abcam). Senescence was induced by treating cells with 100 μM $H_2O_2$ for 4 h.

HUVECs (human umbilical vein cells, CP-H082) were purchased from Procell Life Science&Technology Co. and cultured in DMEM (Gibco, Thermo Fisher Scientific) supplemented with 10% FBS (Gibco, Thermo Fisher Scientific) plus 1% penicillin-streptomycin (Servicebio) at 37 °C in a 5% $CO_2$ humidified incubator. For IL-10 in vitro neutralization, HUVECs were randomized into cytokine-blocking mAb-treatment (IL-10) or HRPN rat-IgG1 groups. 4 μg/ml of HRPN (rIgG1) or IL-10 (JES5-2A5, BioXcell) was administrated for 6 h.

For qRT-PCR, western blot, and tube formation analysis, HUVECs were treated with 50 ng/ml IL-10 (35979S, Cell Signaling Technology), or vehicle, or IL-10 plus Stattic (5 μmol/L, GlpBio) for 24 h.

### ChIP assay

HUVECs were exposed to IL-10 at a concentration of 50 ng/ml (Catalog No. 35979S, Cell Signaling Technology) or vehicle for 24 h subsequent to starvation for this experiment. The Thermo Fisher ChIP Kit (Catalog No. 26156) was utilized following the manufacturer's guidelines. In brief, formaldehyde was introduced to HUVECs to facilitate protein-DNA cross-linking. After lysing the cells in 1.5 ml of lysis buffer (comprising 50 mM HEPES, pH 7.5, 140 mM NaCl; 1 mM EDTA; 1% Triton X-100; 0.1% sodium deoxycholate; 0.1% sodium dodecyl sulfate), genomic DNA fragmentation was achieved by sonication using a Bioruptor ultrasonic cell disruptor (Diagenode). 1% of the sample was set aside to function as an input control. ChIP was conducted according to the provided protocol employing an antibody targeting pSTAT3 (Catalog No. 9131, Cell Signaling Technology). Primer details are provided in Supplementary Table 2.

### qRT-PCR

We extracted total RNA from lumbar spinal endplates or cells by using TRIzol reagent (Invitrogen, Carlsbad, CA) following the manufacturer's protocol. The purity of RNA was assessed by determining the absorbance ratio at 260 nm to 280 nm. Subsequently, the SuperScript First-Strand Synthesis System (Invitrogen) was used to reverse transcribe 1 μg of RNA into cDNA. qRT-PCR was conducted using SYBR Green-Master Mix (Qiagen, Hilden, Germany) on a C1000 Thermal Cycler (Bio-Rad Laboratories, Hercules, CA). Relative gene expression levels were determined using the $2^{-\triangle\triangle CT}$ method. Primer sequences used for qRT-PCR analysis are provided in Supplementary Table 3.TaqMan® Gene Expression Assays.

The Thermo Fisher TaqMan® Gene Expression Assays (catalog no. 4418724 and 4444556) were used according to the manufacturer's instructions. The panel of assays in the TaqMan® Array 96-well Mouse Immune Response Plate targets genes from immune system functions that fall into 9 classes: Cell Surface Receptors; Stress Response;

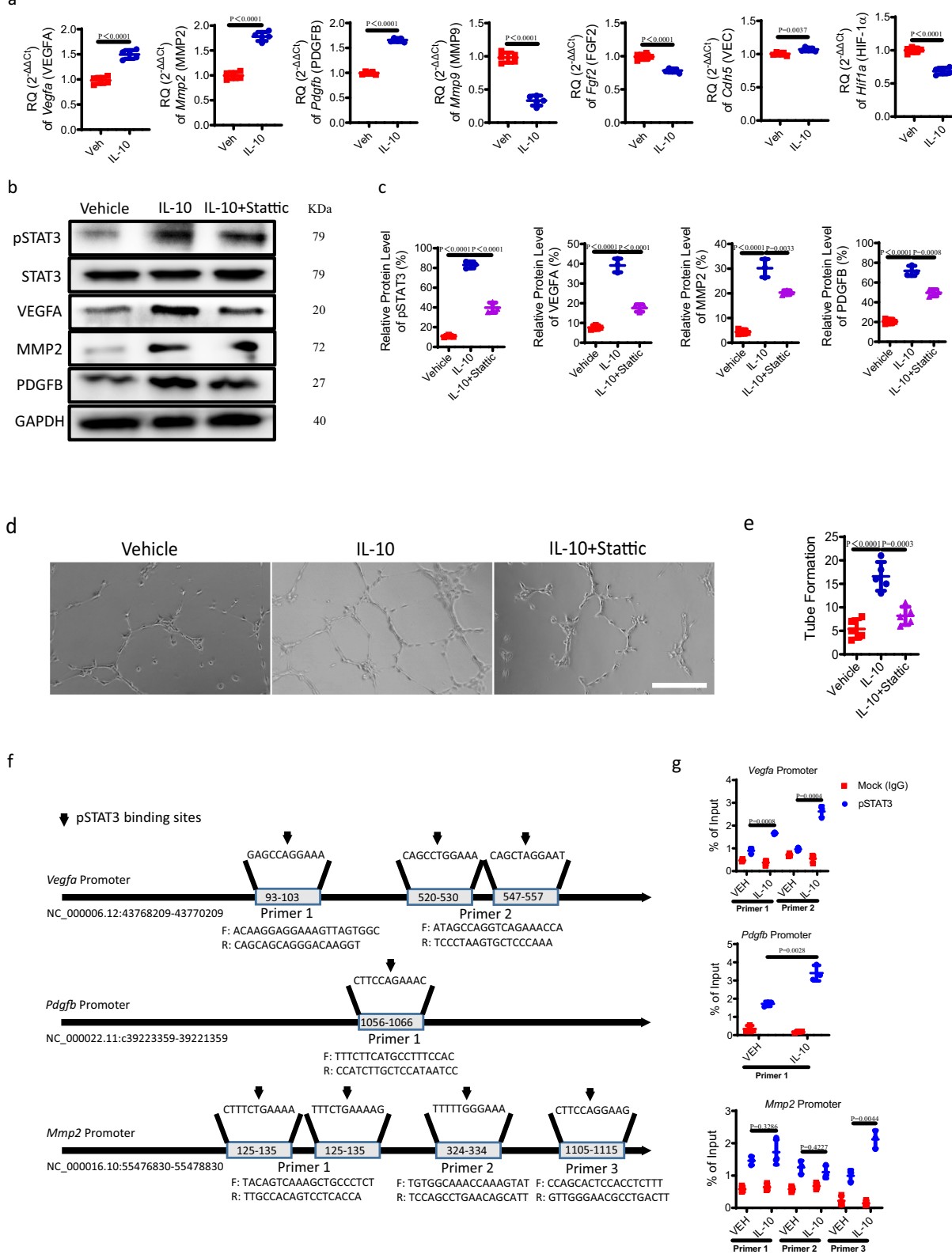

**Fig. 10 | IL-10 stimulates pSTAT3 signaling to induce angiogenesis. a** mRNA expression of several key angiogenesis-associated factors, including Vegfa (VEGFA), Mmp2 (MMP2), Pdgfb (PDGFB), Mmp9 (MMP9), Fgf2 (FGF2), Hif1a (HIF-1α), and Cdh5 (VEC) in endothelial cells treated with 50 ng/ml IL-10 or vehicle for 24 h, determined by qRT-PCR. $n = 5$ per group. **b**–**g** Endothelial cells were treated with 50 ng/ml IL-10 or vehicle for 24 h, plus 5 μmol/L nonpeptidic selective Stat3 inhibitor (Stattic) for 24 h. **b** Western blots of the phosphorylation of STAT3 (pSTAT3), STAT3, VEGFA, MMP2, and PDGFB in endothelial cells. **c** Quantitative analysis of **b**. $n = 3$ per group. **d** Representive tube formation images of endothelial cells. Scale bars, 50 μm. **e** Quantitative analysis of **d**. $n = 5$ per group. **f** Predicated pSTAT3 binding sites on Vegfa, Pdgfb, and Mmp2 promoters and sequences of specific primers. **g** ChIP analysis of pSTAT3 binding on specific Vegfa, Pdgfb, and Mmp2 promoters in endothelial cells with vehicle or IL-10 treatment. $n = 3$ per group. Data are represented as means ± standard deviations, as determined by two-tailed Student's $t$ test or One-way ANOVA. Source data are provided as a Source Data file.

Oxidoreductases; Proteases; Transcription Factors; Signal Transduction; Cytokines & Cytokine Receptors; Chemokines & Chemokine Receptors; and Cell Cycle & Protein Kinases.

## ELISA

The concentrations of IL-10 in the L3–L5 endplates or cells were determined by Mouse IL-10 ELISA Kit (M1000B, R&D Systems) according to the manufacturer's instructions. The concentrations of IL-10 in the human tissues were determined by Human IL-10 ELISA Kit (S1000B, R&D Systems) according to the manufacturer's instructions.

## Tube formation assay

The assay of tube formation was evaluated using the In Vitro Angiogenesis Assay Tube Formation Kit (Cultrex, USA) according to the manufacturer's instructions. In brief, HUVECs were resuspended and seeded in 48-well plates pre-coated with Matrigel in 20% confluency. To visualize tube formation, a BX51 microscope (Olympus) was uesd. The network structure was quantitatively analyzed in five randomly selected fields. Each experiment was repeated five times ($n = 5$).

## Western blot analysis

Cells were lysed in RIPA buffer (Solarbio) plus a phosphatase inhibitor cocktail (Epizyme Biotech). The supernatants of lysates were separated by SDS-PAGE and then blotted on the nitrocellulose blotting membranes (MilliporeSigma, Burkington, MA). The membranes were incubated with primary antibodies against Phospho-STAT3 (1:1000, 9145S, CST), STAT3 (1:1000, 10253-2-AP, Proteintech), MMP2 (1:1000, 10373-2-AP, Proteintech), VEGFA (1:1000, 19003-1-AP, Proteintech), PDGFB (1:1000, AF0240, Affinity) and GAPDH (1:10000, ET1601-4, HUABIO), P16 (1:1000, AF5484, Affinity), P21 (1:1000, 28248-1-AP, Proteintech), γH2A.X (1:1000, ab81299, Abcam), TGFβ (1:1000, sc-130348, Santa Cruz Biotechnology, Inc.), Bax (1:1000, 50599-2-Ig, Proteintech), Bcl2 (1:1000, 3498S, CST), F4/80 (1:500, 14-4801-82, eBioscience), CD31 (1:1000, sc-376764, Santa Cruz Biotechnology, Inc.), OSX (1:1000, A18699, ABclonal) and β actin (1:5000, 20536-1-AP, Proteintech) overnight at 4 °C. Then, the membranes were washed with TBST and incubated with secondary antibodies (1:10000, SA00001-2, Proteintech) for 1 h. The membranes were washed with TBST and then exposed to ECL Chemiluminescent Kit (SQ202, Epizyme Biotech). The signals were quantified using Image-Pro Plus software (Media Cybernetics, Rockville, MD, USA).

## Statistics

All data were processed and analyzed using SPSS, version 26.0, software (IBM Corp.). Data are presented as means ± standard deviations. For comparisons between two groups, unpaired, 2-tailed Student's $t$ tests were used. One-way ANOVA with post hoc tests was used for multiple groups comparisons. For all experiments, $p < 0.05$ was deemed significant. All inclusion/exclusion criteria were predetermined and no samples or animals were ruled out of the analysis. The experiments were randomized, and during experiments and outcome evaluation, the investigators were blinded to assignment. Additionally, each sample was only measured once to avoid duplication of data.

## Reporting summary

Further information on research design is available in the Nature Portfolio Reporting Summary linked to this article.

## Data availability

All relevant data supporting the findings of this study are available within the article and its Supplementary Information file. Source data are provided with this paper.

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

## Acknowledgements
The authors should thank Bo Qin of the First Affiliated Hospital of Zhengzhou University for their technical support. This work was sup-ported by the National Natural Science Foundation of China (Grant Nos. 82002353, S.N.), China Postdoctoral Science Foundation (Grant No. 2020 M682360, S.N.), and Young and Middle-aged Discipline Leader Cultivation Project of He'nan Health (S.N.), National Natural Science Foundation of China (Grant Nos. 82072431, S.N.).

## Author contributions
S.N. and Y.F. designed the experiments. Y.F., W.Z., X.H., M.F., C.S., L.Z., and S.N. performed the experiments and data analysis. Y.F. and S.N. wrote the manuscript. G.P., S.N., and H.Z. provided technical support. S.N. and H.Z. provided material support. G.P. and H.Z. provided advice and comments. S.N. and H.Z. organized and supervised the study.

## Competing interests
The authors declare no competing interests.
