## [Peer Review File · Nature Communications]

Senescent-like Macrophages Mediates Angiogenesis for Endplate Sclerosis via IL-10 secretion in Male MiceREVIEWER COMMENTS

Reviewer #1 (Remarks to the Author):

Fan et, al reported that senescent cells accumulated in the sclerotic endplates of lumbar spine instability or aging mouse model. Particularly, macrophage was identified to undergo senescence in the sclerotic endplates. The senescent macrophages promoted angiogenesis to induce endplate sclerosis by IL-10/pSTAT3 signaling. Overall, the findings in this manuscript are very interesting. The experiments were well designed and results are convincing. The following comments need to be addressed to improve the manuscript.

1. Endplate sclerosis is a notable aspect of spine degeneration or aging and a source of spinal pain. The authors demonstrated that the conditional knockout of *cdkn2a* in *Lyz2+* cells could inhibit the innervation of CGRP+ sensory nerves. I am wondering if the senolytic drugs, D+Q could also ablation the sensory innervation. Please clarify it at least in one of the spinal degeneration models, for example immunofluorescent staining of CGRP in lumbar spine instability model.
2. Please include the gating strategy for the flow cytometry analysis in panel E of Figure 5.
3. The authors' study and the literature suggest that sclerosis or lesion of the cartilage endplate correlates with low back pain in patients, whether the number of senescent macrophages within the endplates of human sample in supplementary figure 2 correlates with clinical pain scores?
4. Although the role of cellular senescence in endplate degeneration has not been reported in the literature, it has been documented that senolytic deletion of senescent cells is beneficial for the repair of nucleus pulposus during disc degeneration. This aspect of the presentation should be added to the Introduction part with relevant citation (Novais EJ, Tran VA , Johnston SN, Darris KR, Roupas AJ, Sessions GA, Shapiro IM, Diekman BO, Risbud MV. Long-term treatment with senolytic drugs Dasatinib and Quercetin ameliorates age-dependent intervertebral disc degeneration in mice. Nat Commun. 2021 Sep 3;12(1):5213. doi: 10.1038/s41467-021-25453-2).

5. A growing number of publications have recently reported an important beneficial role for senescent cells, showing that the role of cellular senescence is dual-sided. This should be included in the Discussion part with proper citations, for example: Reyes NS, Krasilnikov M, Allen NC, et al. Sentinel p16INK4a+ cells in the basement membrane form a reparative niche in the lung. *science*. 2022;378(6616):192-201. doi:10.1126/science.abf3326.

6. There are some typos in this manuscript, including but not limited to the below: “identified” in Abstract, “supraspinous ligaments” in Animals and treatment, and “significant” in Statistics.

7. The title the coupling in the title should be changed as “Senescent Macrophages Mediate angiogenesis for endplate sclerosis by secreting IL-10 as SASP Factor.”

Reviewer #2 (Remarks to the Author):

In this manuscript, Fan et al. described the accumulation of senescent cells in the sclerotic endplates of lumbar spine instability (LSI) or aging mouse models. They demonstrated the presence of vessel formation coupled with endplate sclerosis in LSI or aged mice using CD31/Emcn staining. Furthermore, they provided evidence that the accumulation of senescent cells initiates endplate sclerosis. The authors also identified macrophage accumulation and showed that targeting senescent markers can alleviate endplate sclerosis. Finally, they proposed that the secretion of IL-10 from macrophages promotes the activation of cytokines and growth factors, including pSTAT, leading to angiogenesis. They correlated macrophages with angiogenesis and sclerosis in degenerative endplates.

Overall, these findings are very interesting and novel. The manuscript is well-planned and executed. The data is robust and clear and effectively supports the concept.

Comments:

The authors aimed to target senescence to explore endplate sclerosis. They demonstrated

that macrophages influence the angiogenic cytokine IL-10, which contributes to endplate sclerosis. In Figure 1, the CD31+Emcn+ vessels were significantly increased in LSI-4/8 weeks and aged mice, while no significant vessel formation was observed in sham or young mice. It would be valuable to investigate whether nucleus pulposus cells participate in vessel formation, considering their exacerbating factor in intervertebral disc degeneration (IVDD). Additionally, does the composition of the cartilage endplate play a role in vascularity during LSI? Moreover, how did the authors compare sham with 4 W/8 W old mice, as the 3M old mice did not show any evidence of vessel formation?

Figure 2 demonstrated the accumulation of senescent cells in sclerotic endplates, which is impressive, particularly by using the P16 senescence reporter strain. Uncoupling of bone resorption and bone formation is a prime reason for age-related bone loss, which may induce apoptosis in osteocytes and other niche cells. It would be interesting to explore whether the senescence in macrophages results from resistance to apoptosis. Additionally, the authors did not include any apoptotic data analysis while targeting senescence.

Figure 3 showed the inhibition of senescence using a senolytic cocktail (D+Q). The D+Q treatment resulted in a reduction of CD31+/Emcn+ expression in LSI 8 weeks. The study showed that D+Q does not affect P16 expression. It would be helpful to understand how the authors correlated P16 expression with senolytic treatment while eliminating senescent cells. Furthermore, LSI 4 weeks data is missing for comparative analysis.

Figure 5 presented the immunostaining of P16 tdTom with F4/80 cells, suggesting the important role of macrophages in endplate sclerosis. The flow data does not correlate with the imaging data in Figure 5a. The LSI 8 weeks data is missing, which would provide insights into the prolonged expression of F4/80. Additionally, the authors could consider showing the mitochondrial expression for age-related senescence in 20M old mice.

Fig 7 and 8 explained the deletion of macrophage to inhibit LSI or aging induced endplate sclerosis. 8 weeks LSI abrogate the expression of F4/80 and CD31 after DT injection. Administration of CL used to deplete macrophages and this effect was confirmed with F4/80 immunostaining. However data did not show any DT injection to deplete the macrophages

as is mentioned in the text (result section). Please check whether it was required or not. The spatiotemporal effect of DT injection was not shown in 4 weeks LSI.

IL-10 expression was outlined compared to rest of the gene expression in tdTom+F4/80+ senescent macrophages and sourced as the primary responsible cells. It is well established that IL-10 could induce the senescence in HSCs and enhance the expression of senescence markers P53 and P21. How did the author compare the base expression of IL-10 in comparison with macrophages? Alone expression of IL-10 cannot be the only factor for the angiogenic and sclerotic condition in endplate degeneration from macrophages. Author may include additional investigation of angiogenic factors/ cell surface markers to justify the claim.

The important reference on skeletal angiogenesis is not cited in the manuscript as reviewed here: PMID: 34281770, PMID: 32845550.

Reviewer #3 (Remarks to the Author):

In this study, the authors analyzed the molecular mechanism behind endplate sclerosis using both LSI and aging animal models. One of the key findings of the study was the identification of an accumulation of senescent macrophages within the endplates in these models. The authors further stated that senescent macrophages led to an increase in the production of IL-10, an anti-inflammatory cytokine, which promoted angiogenesis through the activation of pSTAT3 and its downstream growth factors within the endplates, contributing to the development of sclerosis. However, several important questions remain unanswered. The following comments highlight the specific areas that require further investigation and clarification.

1. It is unclear where those macrophages originate from and what types of macrophages are present in the endplates. Additionally, the study does not provide convincing information about the percentage of macrophages undergoing senescence or if they are indeed senescent macrophages. Furthermore, it is uncertain whether these senescent macrophages are responsible for the production of anti-inflammatory IL-10.

a) As shown in Figure 5, the control groups and 3 m.o. old young mice have very few resident macrophages (F4/80+) in the endplate. Therefore, the increased presence of F4/80+ macrophages observed in the LSI groups or 20 m.o. old mice is likely due to infiltrated macrophages. It is necessary to analyze the types of infiltrated macrophages, such as M1 or M2, through co-staining with markers like CD206, CD163, and others. Additionally, co-staining of F4/80 with CD68, CD11b, and/or CD45 should be performed to confirm that these infiltrated cells are indeed macrophages.

b) It is not logical to assume that macrophages are already senescent upon infiltration into the endplate. Rather, macrophages may undergo senescence after infiltration. To address this question, the authors should analyze when macrophages infiltrate the endplate and when they start expressing senescence markers. This can be achieved by analyzing sequential time points, such as examining 6, 12, and 16 m.o. old mice in addition to the 3 m.o. and 20 m.o. old mice used in the aging models.

c) Diagnosing macrophage senescence is not straightforward, as even non-senescent macrophages can express markers like p16 and SA- β -gal. Based on Figure 5 (E, F), around 40% of macrophages in the control (sham) mice are double positive for F4/80 and tdTomato (about 60% in the LSI group). These 40% of double positive macrophages found in the control young mice are unlikely to be senescent macrophages. Therefore, at most, 20% of macrophages in the LSI group could be considered "senescent". However, it remains uncertain whether these 20% of "senescent macrophages" are truly senescent and whether they are responsible for the elevation of IL-10 expression and the phenotypic changes observed in the LSI models. Thus, the title of the manuscript, "Senescent macrophages mediates angiogenesis --- by IL-10", is not justified at this stage.

d) Senescent cells typically secrete pro-inflammatory factors as part of their SASP, whereas IL-10 is an anti-inflammatory cytokine. Therefore, it is important to consider the possibility of M2-like macrophages, rather than "senescent macrophages", being involved in the increase of IL-10 in the endplates of LSI and aging models. The subtypes of infiltrated macrophages (and/or resident macrophages) and cytokines including IL-10 should be verified through co-staining etc., as mentioned above.

e) Furthermore, the authors state that "the most specific markers for cellular senescence are SADS and gH2A.X" (page 8). However, this is an overstatement and incorrect. Additionally, in Figures 5H and 5J, SADS+/F4/80+ or gH2A.X+/F4/80+ double positive cells

should be normalized to the total number of F4/80+ cells rather than per area.

2. The authors knocked out p16 in macrophages and granulocytes and observed that p16 knockout prevents endplate sclerosis in the LSI models. p16 depletion only will not reverse the state of senescence, therefore the striking phenotypic changes found in the KO mice are somewhat surprising. The authors should at least analyze the expression levels of cytokines, particularly IL-10, using Q-PCR and determine the cells expressing IL-10 by performing co-immunostaining of IL-10 with p16, gH2A.X, F4/80, CD206 etc to see if IL-10 acts as a downstream mediator of p16 in the development of sclerosis.

3. The phenotypic changes observed in the iDTR system suggest the involvement of macrophages in sclerosis. However, it is crucial to analyze the expression changes of cytokines such as IL-10, IL-1 β , TNF in the macrophage/granulocyte-depleted endplates.

4. About the clearance of senescent cells using D+Q: It should be noted that current senolytic treatments, including D+Q, are not specific to senescent cells. To evaluate the effect of senescent cells on sclerosis, additional senolytics that act through different mechanisms, such as ABT737, should be considered.

5. In most figures, the data heavily rely on histological analyses. Strengthening the data by including Western blotting and Q-PCR for proteins/mRNAs like CD31, Emcn, OSX, OCN, HMGB1, F4/80, tdTomato, among others, would provide additional support.

6. TGF β has been reported to promote endplate sclerosis (Scientific Reports | 6:27093 | DOI: 10.1038/srep27093). TGF β is a SASP factor, and both IL-10 and TGF- β can be expressed by M2 macrophages. Therefore, it is essential to examine the expression levels of TGF β in the models investigated in this study.

Reviewer #4 (Remarks to the Author):

Fang and colleagues have examined the role of senescent macrophages in endplate sclerosis using multiple animal models and pharmacological approaches. They gathered data to

suggest that inhibition of vegf or IL10, deletion of p16 expression in cells of the macrophage lineage, deletion of macrophages, or administration of the senolytics desatinib and quercetin all inhibit endplate sclerosis. From these results they conclude that IL10 produced by senescent macrophages contributes to endplate vascularization. This idea is novel but conceptually it is difficult to understand. This is because the presence of macrophages in chondrocytic tissues is dependent on blood vessel invasion. In addition, the amount of CD31+Emcn+ blood vessels present at 4 and 8 wk after LSI surgery or with aging is directly dependent on the size of the pores created by the osteoclasts, as previously shown by the authors. However, for unknown reasons, in this manuscript the authors do not provide any information about osteoclasts. Because osteoclasts are essential for the enlargement of the pores and mineralization of the endplates, it is likely that changes in osteoclasts are a critical determinant of the changes in porosity and sclerosis with the different interventions. Without the osteoclast component it is difficult to envision how the mechanism proposed by the authors works. Moreover, because the authors analyze the effects of their genetic or therapeutic intervention at time points when the endplate sclerosis is well developed, it is not possible to pin down the cascade of events that contribute to the effects measured. Another problem with the work presented is that even under normal physiological circumstances, β -galactosidase activity is enriched in particular cell types, such as mature tissue macrophages and osteoclasts. Moreover, in macrophages, high p16Ink4a expression is not directly associated with senescence. (Odgren et al. *Connect Tissue Res*, 2006; Kopp et al. *Histol Histopathol*, 2007; Cudejko et al. *Blood*, 2011; Hall et al. *Aging*, 2017). In many instances there is not enough detail provided in the methods for the work to be reproduced and some of the assays are not described. Overall, the conclusions of this work are not well supported by the data provided. The following are a list of other concerns raised by the work presented.

Major points

- 1- The authors use several markers to indicate endplate sclerosis, such as porosity, endplate scores, area of CD31+Emcn+ blood vessels, and number of Osx and Ocn positive cells. However, these markers are redundant and their changes are correlated because the amount of vessels and osteoblastic cells depends on the number and size of the pores. Moreover, the endplate scores are not defined in the methods.
- 2- The findings that CD31+EMCN+ blood vessels are present in the porous endplates after

LSI surgery and aging (Fig. 1) has been previously shown by the authors.

3- In figure 2A and H, the authors should show a picture of the whole intervertebral area so that the extent of the b-gal staining can be appreciated.

4- In Figure 2 D-E and K-L the comparison between HMGB1 in chondrocytes versus some unidentified cells in the pores is meaningless.

5- In figure 2 M-N, the fold change in expression most of the genes analyzed is about 4. This is statistically very improbable. The authors should confirm whether this is indeed the case and provide the gene expression without the normalization of the control values to 1 in these experiments and others throughout the manuscript.

6- In some of the images the immunohistochemistry for Osx and OCN is overstained because all the cells are brown, including some of the chondrocytes seen in the high magnification images. In any case, the significance of the changes seen in the number of these cells is unclear, other than just a reflection of the cellular content of the pores. For example, the authors claim that administration of D+Q to aged mice (Fig 4J-K) decreases the number of Osx cells within the pores. How would this contribute to the decrease in porosity by D+Q? In contrast to this findings, D+Q increases osteoblastic cells in aged bone (Farr et al, Nature Medicine 2017).

7- In figure 5E-F, the quantification of Tomato positive macrophages indicates that 40% of macrophages are positive for p16 in sham operated mice. The origin of these macrophages is unclear because the histology images do not reveal the presence of macrophages in the endplates of sham operated mice. In any case, this is in agreement with multiple publications indicating that macrophages express p16 independent of senescence. However, in the other panels of figure 5 the authors quantified senescence markers in sham mice at sites where there are no macrophages present and therefore all readings in sham mice are zero. Thus, all these graphs and images merely indicate the presence or absence of macrophages. Does the deletion of p16 or D+Q decrease the presence of SADS or DNA damage markers in macrophages?

8- Is there any potential reason for why the macrophages become senescent in the pores of mice with LSI surgery?

9- In figure 6, the authors should quantify the efficacy of deletion of p16, even if in macrophages from the bone marrow. Do p16 deleted macrophages become less senescent and produce less IL10? Does p16 deletion alter osteoclast number?

10- It is unclear how 1 month of administration of D+Q to old mice, in which endplate sclerosis is established, can significantly decrease porosity and seemingly rejuvenate the cartilaginous portion. This is in stark contrast to the findings of Novais et al (Nat Commun 2021) showing that administration of D+Q to 18-month-old mice with established intervertebral disk and endplate degeneration had no effect.

11- In the experiments in which macrophages are deleted (Fig 7 and 8) the number of osteoclasts must be directly impacted, as macrophages are the precursors of osteoclasts. However, this is not even considered in the discussion of these experiments. The relevance of this work is unclear.

12- The experiment of fig. 9A is not described in the methods or in the figure legend. The figure is not labeled appropriately.

13- In Fig 10C, the majority of macrophages expresses Tomato. However, senescence is a cell fate that affects a small minority of cells in vivo. Thus, it is unlikely that these markers reflect cellular senescence. In the image provided there are also a number of cells other than macrophages that express IL10. The background of the two panels in figure 10E is very different and therefore is difficult to appreciate any differences. In fig 10F, the expression of *cdkn2a* should be measured.

Minor points

1- The labels of the Y axis of the graphs showing CD31+EMCN+ blood vessels are confusing because the graph shows %0 and the legends show percent.

2- The sex and background strain of the mice are not described.

3- Many measurements are reported per area (/Ar) but these should be clarified.

4- There are many typos throughout the manuscript.

Reviewer #1 (Remarks to the Author):

Fan et, al reported that senescent cells accumulated in the sclerotic endplates of lumbar spine instability or aging mouse model. Particularly, macrophage was identified to undergo senescence in the sclerotic endplates. The senescent macrophages promoted angiogenesis to induce endplate sclerosis by IL-10/pSTAT3 signaling. Overall, the findings in this manuscript are very interesting. The experiments were well designed and results are convincing. The following comments need to be addressed to improve the manuscript.

Response: We are encouraged by the reviewer's overall insightful comments.

1. Endplate sclerosis is a notable aspect of spine degeneration or aging and a source of spinal pain. The authors demonstrated that the conditional knockout of *cdkn2a* in *Lyz2*⁺ cells could inhibit the innervation of CGRP⁺ sensory nerves. I am wondering if the senolytic drugs, D+Q could also ablation the sensory innervation. Please clarify it at least in one of the spinal degeneration models, for example immunofluorescent staining of CGRP in lumbar spine instability model.

Response: We appreciate the reviewer's valuable suggestion. As suggested by the reviewer, we added the detection of sensory nerves in the LSI model with D+Q treatment. The immunofluorescent staining demonstrated that the administration of D+Q significantly decreased the innervation of CGRP⁺ sensory nerve in the LSI mice. We included it the Supplementary Figure 12 in the revised manuscript.

2. Please include the gating strategy for the flow cytometry analysis in panel E of Figure 5.

Response: We appreciate the reviewer's important suggestion. We have added the gating strategy for the flow cytometry analysis in the revised manuscript as Supplementary Figure 5.

3. The authors' study and the literature suggest that sclerosis or lesion of the cartilage endplate correlates with low back pain in patients, whether the number of senescent macrophages within the endplates of human sample in supplementary figure 2 correlates with clinical pain scores?

Response: We appreciate the reviewer's important suggestion. As we mentioned earlier, we found that the sensory innervation was significantly relieved in the endplate by the elimination of senescent cells. Therefore, as suggested by the reviewer, exploring the relationship between the presence of senescent cells and pain is a very valuable suggestion. We retrospectively analyzed the correlation between the preoperative Visual Analogue Scale (VAS) pain score and the number of P16⁺CD68⁺ cells in the endplate tissue. We found a positive correlation between VAS pain score and the number of P16⁺CD68⁺ cells, as shown in Supplementary Figure 19e in the revised manuscript. This is consistent with the coupling process initiated by senescence cells, as confirmed by our research demonstration.

4. Although the role of cellular senescence in endplate degeneration has not been reported in the literature, it has been documented that senolytic deletion of senescent cells is beneficial for the repair of nucleus pulposus during disc degeneration. This aspect of the presentation should be added to the Introduction part with relevant citation (Novais EJ, Tran VA, Johnston SN, Darris KR, Roupas AJ, Sessions GA, Shapiro IM, Diekman BO, Risbud MV. Long-term treatment with senolytic drugs Dasatinib and Quercetin ameliorates age-dependent intervertebral disc degeneration in mice. *Nat Commun.* 2021 Sep 3;12(1):5213. doi: 10.1038/s41467-021-25453-2).

Response: We appreciate the reviewer's important suggestion. We have added this reference and stated as "The senolytic drugs could delay the age-related intervertebral disc degeneration" in the revised manuscript

5. A growing number of publications have recently reported an important beneficial role for senescent cells, showing that the role of cellular senescence is dual-sided. This should be included in the Discussion part with proper citations, for example: Reyes NS, Krasilnikov M, Allen NC, et al. Sentinel p16INK4a+ cells in the basement membrane form a reparative niche in the lung. *science.* 2022;378(6616):192-201. doi:10.1126/science.abf3326.

Response: We appreciate the reviewer's important suggestion. We have added this valuable reference in the revised manuscript as "It has been recently reported an important beneficial role for senescent cells in epithelial regeneration, showing that the role of cellular senescence is dual-sided".

6. There are some typos in this manuscript, including but not limited to the below: "identified" in Abstract, "supraspinous ligaments" in Animals and treatment, and "significant" in Statistics.

Response: We appreciate the reviewer's comments. We feel sorry for the typos. The typos have been corrected in the revised manuscript.

7. The title the coupling in the title should be changed as "Senescent Macrophages Mediate angiogenesis for endplate sclerosis by secreting IL-10 as SASP Factor."

Response: We appreciate the reviewer's excellent suggestion. We have revised the title as "Senescent Macrophage Mediates Angiogenesis for Endplate Sclerosis by Secreting IL-10 as SASP in Male Mice"

Reviewer #2 (Remarks to the Author):

In this manuscript, Fan et al. described the accumulation of senescent cells in the sclerotic endplates of lumbar spine instability (LSI) or aging mouse models. They demonstrated the presence of vessel formation coupled with endplate sclerosis in LSI or aged mice using CD31/Emcn staining. Furthermore, they provided evidence that the accumulation of senescent cells initiates endplate sclerosis. The authors also identified macrophage accumulation and showed that targeting senescent markers can alleviate endplate sclerosis. Finally, they proposed that the secretion of IL-10 from macrophages promotes the activation of cytokines and growth factors, including pSTAT, leading to angiogenesis. They correlated macrophages with angiogenesis and sclerosis in degenerative endplates.

Overall, these findings are very interesting and novel. The manuscript is well-planned and executed. The data is robust and clear and effectively supports the concept.

Response: We are encouraged by the reviewer's overall insightful comments.

Comments:

1. The authors aimed to target senescence to explore endplate sclerosis. They demonstrated that macrophages influence the angiogenic cytokine IL-10, which contributes to endplate sclerosis. In Figure 1, the CD31+Emcn+ vessels were significantly increased in LSI-4/8 weeks and aged mice, while no significant vessel formation was observed in sham or young mice. It would be valuable to investigate whether nucleus pulposus cells participate in vessel formation, considering their exacerbating factor in intervertebral disc degeneration (IVDD). Additionally, does the composition of the cartilage endplate play a role in vascularity during LSI? Moreover, how did the authors compare sham with 4 W/8 W old mice, as the 3M old mice did not show any evidence of vessel formation?

Response: We appreciate the reviewer's valuable suggestion. To investigate whether nucleus pulposus cells participate in vessel formation in degenerative endplates, we carefully reviewed the literature. The vascularization is noted in the degenerated discs (PMID: 23736847, PMID: 25209447, PMID: 28970490). The expression level of VEGF-A and CD31 may increase in degenerative discs of patients, which could trigger angiogenesis (PMID: 37626681). Additionally, the level of angiogenic factor angiopoietin-2 increased in degenerative discs in patients (PMID: 28394321). Among the nucleus pulposus cells (NPCs), the fibroNPCs showed the highest score of angiogenesis compared to other types of NPC, as evidenced by single-cell Transcriptome Profiling (PMID: 34825784). We re-analyzed the immunostaining of CD31⁺Emcn⁺ in nucleus pulposus tissue of LSI-8 weeks mice. We did not observe obvious angiogenesis in the nucleus pulposus tissue (see below). Together, the senescent macrophage-derived IL-10 played a key role in the angiogenesis in

degenerative endplates, but the angiogenic factors from NPCs might also be involved in the angiogenesis process. We have included this in the discussion part in the revised manuscript.

Usually, cartilage is a physiologically avascular tissue, but some angiogenic factors might be released from the ECM and be activated to participate in the angiogenesis process, such as TGF- β 1 (PMID: 32945489). Although the administration of IL-10 neutralizing antibody demonstrated that IL-10 was crucial to induce angiogenesis in sclerotic endplates in this study, the composition of degenerative cartilage endplates may also be involved in the angiogenesis process. We have included this in the discussion part in the revised manuscript.

Our grouping design was to compare the LSI surgery mice with the Sham surgery mice. For LSI group, we collected the samples at 4 weeks or 8 weeks post surgery. For the aging model, 3 month old mice are selected as young mice and 20 month old mice are selected as aging mice for further analysis.

2. Figure 2 demonstrated the accumulation of senescent cells in sclerotic endplates, which is impressive, particularly by using the P16 senescence reporter strain. Uncoupling of bone resorption and bone formation is a prime reason for age-related bone loss, which may induce apoptosis in osteocytes and other niche cells. It would be interesting to explore whether the senescence in macrophages results from resistance to apoptosis. Additionally, the authors did not include any apoptotic data analysis while targeting senescence.

Response: We appreciate the reviewer's important suggestion. Recent research has shown that senescent cells exploit anti-apoptotic machinery to survive (PMID: 31746100, PMID: 25754370, PMID: 31838837). The senolytic we used eliminate senescent cells by inhibiting the anti-apoptotic proteins. Based on our results of SA- β -Gal staining in Fig. 3B and Fig. 4B, many senescent cells in porous endplates are eliminated by senolytic. Furthermore, we conducted the western blot of the anti-apoptotic Bcl-2 proteins and pro-apoptotic protein Bax on the senescent bone marrow-derived macrophages. The results demonstrated that the senescence in macrophages is accompanied by an increase in anti-apoptotic activity (Supplementary Figure 9 in the revised manuscript).

3. Figure 3 showed the inhibition of senescence using a senolytic cocktail (D+Q). The D+Q treatment resulted in a reduction of CD31⁺/Emcn⁺ expression in LSI 8 weeks. The study showed that D+Q does not affect P16 expression. It would be helpful to understand how the authors correlated P16 expression with senolytic treatment while eliminating senescent cells. Furthermore, LSI 4 weeks data is missing for comparative analysis.

Response: We appreciate the reviewer's comment. SA-bgal and P16 are the markers of cellular senescence. The LSI surgery or aging model could induce cellular senescence in sclerotic endplates, as indicated by the accumulation of P16 and SA-bgal. Considering the similar indicative role of P16 and SA-bgal, we chose the SA-bgal staining to evaluate the treatment of senolytic cocktail. Here, the immunostaining of tdTom (P16) was added in Figure 3d, e in the revised manuscript. The data also showed that the D+Q treatment significantly decreased the number of tdTom⁺ cells. Additionally, to investigate the effect of D+Q treatment on the LSI-4W mice, the staining of SA-bgal, CD31/Emcn, and Safranin O Fast Green was included in Supplementary Figure 2 of the revised manuscript.

4. Figure 5 presented the immunostaining of P16 tdtom with F4/80 cells, suggesting the important role of macrophages in endplate sclerosis. The flow data does not correlate with the imaging data in Figure 5a. The LSI 8 weeks data is missing, which would provide insights into the prolonged expression of F4/80. Additionally, the authors could consider showing the mitochondrial expression for age-related senescence in 20M old mice

Response: We appreciate the reviewer's insightful comments. For histological sectioning and staining, we conducted coronal sections of the spinal tissues. To ensure consistency of comparison in different samples, we chose sections in the middle part for staining and analysis. In the histological sections, we could clearly distinguish the boundary between the endplate tissue and the nucleus pulposus or vertebral body tissue. However in the flow cytometry experiments, although we used a microscope to separate the endplate tissue, it was difficult to ensure that there was no mixed nucleus pulposus or vertebral body tissue. We repeated the flow cytometry analysis in an attempt to explore the potential reason for the detection of F4/80⁺ cells in the control group. Against expectations, F4/80⁺ cells were still detected in the sham group in the flow cytometry data. However, the number of tdTom⁺F4/80⁺ cells in the sham group was significantly lower than in the LSI group. In addition, during the preparation of single-cell suspensions of endplate tissues, the digestion time was longer because the endplates were cartilage or bone tissues. Although we shortened the digestion time when repeating the experiment, it still inevitably affected the cell state, which may also contribute to the presence of a certain number of tdTom⁺F4/80⁺ cells in the sham group. Together, these limitations do not affect the results that the number of tdTom⁺F4/80⁺ cells increased significantly in the LSI group relative to the sham group, as indicated by flow cytometry data. We have included more representative data of flow cytometry in Figure 5e in the revised manuscript

As the reviewer's suggestion, we have added the immunofluorescent staining of F4/80

with tdTom in the endplate at 8 weeks after LSI surgery in the revised manuscript as Supplementary Figure 4. The data showed that the number of tdTom⁺F4/80⁺ cells was significantly higher in the endplates at 8 weeks after LSI surgery relative to sham surgery.

Mitochondrial dysfunction and cell senescence are hallmarks of aging. Cell senescence was defined as the irreversible cell cycle arrest, initiated as a persistent DNA damage and activation of the senescence-associated secretory phenotype (SASP). Mitochondrial dysfunction is a decrease in respiratory capacity with low mitochondrial membrane potential (MMP). Usually, the consequences of mitochondrial dysfunction include decreased mitophagy, increased mitochondrial mass, loss of mitochondrial function, and increased production of reactive oxygen species (ROS). Although mitochondrial dysfunction and cell senescence are different cellular states, they are closely interconnected. Senescence could drive mitochondrial dysfunction. For example, 1) mitochondrial DNA (mtDNA) damage could cause mitochondrial dysfunction, 2) mitochondria of senescent cells undergo structural changes associated with increases in size and volume, 3) dysregulated nutrient sensing pathways (such as MAPK, SIRT, and mTOR) in senescence could induce mitochondrial dysfunction, 4) activation of IP3P led to Ca²⁺ release to induce mitochondrial Ca²⁺ overload. Meanwhile, mitochondrial dysfunction governs the senescent phenotype. For example, 1) impaired mitochondria produce excessive ROS to cause DNA damage and cell cycle arrest, 2) reduced NAD⁺/NADH phosphorylates and/or stabilizes p53 and p16INK4 mRNA, 3) dysfunctional mitochondria contribute to innate immune response activation to produce inflammation, 4) multiple antiapoptotic pathways are upregulated in senescence, presumably in response to partial mitochondrial membrane permeabilization. For details, a recent review is referred (PMID: 35775483). In the present study, we focused on the role of senescent macrophages in LSI or aging-induced endplate sclerosis. The mitochondrial dysfunction may contribute to the senescence during endplate degeneration. We have included this part in the Discussion in the revised manuscript.

5. Fig 7 and 8 explained the deletion of macrophage to inhibit LSI or aging induced endplate sclerosis. 8 weeks LSI abrogate the expression of F4/80 and CD31 after DT injection. Administration of CL used to deplete macrophages and this effect was confirmed with F4/80 immunostaining. However data did not show any DT injection to deplete the macrophages as is mentioned in the text (result section). Please check whether it was required or not. The spatiotemporal effect of DT injection was not shown in 4 weeks LSI.

Response: We appreciate the reviewer's comment. For the LSI model, we employed *iDTR^{Allyz2}* mice, in which deletion of Lyz2-positive cells could be induced by injection of DT. The knockout efficiency of macrophages by DT was verified by immunofluorescent staining of F4/80, as shown in the Figure 7A. For the aging model, CL was used to eliminate macrophages. The knockout efficiency of macrophages by CL was verified by immunofluorescent staining of F4/80 in the Figure 8A. We apologize for the typos and unclear statement.

Macrophage senescence and angiogenesis of CD31⁺/Emcn⁺ blood vessels occurred at 4W and 8W after LSI surgery compared with sham surgery. Importantly, the cumulative number of senescent macrophages and the number of neovascularizations are higher at 8W after LSI surgery than that at 4W after LSI surgery. Therefore, we chose the time point of 8W after LSI surgery to evaluate the effect of macrophage clearance on LSI-induced endplate degeneration. As the reviewer's comment, to show the spatiotemporal effect of DT injection, we added the Safranin O and Fast Green staining of spinal sections from *iDTR^{ΔLy2}* mice at 4W after LSI surgery with DT injection in Supplementary Figure 13 of the revised manuscript.

6. IL-10 expression was outlined compared to rest of the gene expression in tdTom⁺F4/80⁺ senescent macrophages and sourced as the primary responsible cells. It is well established that IL-10 could induce the senescence in HSCs and enhance the expression of senescence markers P53 and P21. How did the author compare the base expression of IL-10 in comparison with macrophages? Alone expression of IL-10 cannot be the only factor for the angiogenic and sclerotic condition in endplate degeneration from macrophages. Author may include additional investigation of angiogenic factors/ cell surface markers to justify the claim.

Response: We appreciate the reviewer's valuable suggestion. To explore the potential key source to induce angiogenesis during endplate sclerosis, we first analyzed target genes from immune system functions that fall into 9 classes: Cell Surface Receptors, Stress Response, Oxidoreductases, Proteases, Transcription Factors, Signal Transduction, Cytokines & Cytokine Receptors, Chemokines & Chemokine Receptors, and Cell Cycle & Protein Kinases using a Mouse Immune Response Plate. As shown in Figure 9, IL-10 was the most elevated gene in the endplates of 20-month-old mice (Figure. 9a). The increased protein level of IL-10 in the endplates of aged mice was validated by enzyme-linked immunoabsorbent assay (ELISA) (Figure. 9b). Immunofluorescent staining further demonstrated that tdTom⁺F4/80⁺ senescent macrophages were the source of IL-10 secretion in sclerotic endplates (Figure. 9c). Further, the in vitro experiments showed that the release of IL-10 from senescent macrophages increased significantly (figure. 9e, f). Although the administration of 2A5, an anti-IL-10 neutralizing antibody validated the key function of IL-10 in angiogenesis in vivo and in vitro, we strongly agree with the reviewer's comment that IL-10 cannot be the only factor for the angiogenic and sclerotic condition in endplate degeneration. As indicated by our data in Figure 9a and 9f, the expression of CCL2 was also elevated in degenerative endplates and senescent macrophages, which could promote angiogenesis via activating Ets-1 (PMID: 16888027) and VEGF (PMID: 21515678). It has been reported that the dysregulation of ATP binding cassette transporter ABCA1 in senescent macrophages may contribute to pathological angiogenesis (PMID: 23562078). Additionally, The TGF-β has been reported to be involved in endplate sclerosis (PMID: 27256073). We conducted the co-immunofluorescent staining of TGF-β with F4/80 in vivo in Supplementary Figure 18a of the revised manuscript. The data showed that the TGF-β could co-localize with F4/80 in the endplates of LSI group or aged group. Additionally, the western blot also showed that

the total level of TGF- β in the senescent macrophage increased in vitro relative to naive macrophage, as revealed in Supplementary Figure 18b,c of the revised manuscript. However, as the literature reported, the latent TGF- β was mainly stored in the extracellular matrix and then released and activated in responding to stimuli (PMID: 24487640). Similarly, the level of active TGF- β increased in the endplates at 2W, 4W, and 8W post LSI surgery, but the total TGF- β only increased in the endplates at 2W post LSI surgery and then returned to basal level at 4W and 8W post LSI surgery (PMID: 27256073). Therefore, the activation of TGF- β is more important than the total TGF- β . Together, these indicate that the TGF- β released by senescent macrophage might participate in the endplate sclerosis, but may not be the most important source of active TGF- β . We have included it in the discussion part of the revised manuscript.

7. The important reference on skeletal angiogenesis is not cited in the manuscript as reviewed here: PMID: 34281770, PMID: 32845550.

Response: We appreciate the reviewer's important suggestion. We have added these valuable literature to the revised manuscript.

Reviewer #3 (Remarks to the Author):

In this study, the authors analyzed the molecular mechanism behind endplate sclerosis using both LSI and aging animal models. One of the key findings of the study was the identification of an accumulation of senescent macrophages within the endplates in these models. The authors further stated that senescent macrophages led to an increase in the production of IL-10, an anti-inflammatory cytokine, which promoted angiogenesis through the activation of pSTAT3 and its downstream growth factors within the endplates, contributing to the development of sclerosis. However, several important questions remain unanswered. The following comments highlight the specific areas that require further investigation and clarification.

Response: We appreciate the reviewer's overall insightful and accurate comments. We have addressed all the comments with additional experimentation and clarification.

1. It is unclear where those macrophages originate from and what types of macrophages are present in the endplates. Additionally, the study does not provide convincing information about the percentage of macrophages undergoing senescence or if they are indeed senescent macrophages. Furthermore, it is uncertain whether these senescent macrophages are responsible for the production of anti-inflammatory IL-10.

a) As shown in Figure 5, the control groups and 3 m.o. old young mice have very few resident macrophages (F4/80+) in the endplate. Therefore, the increased presence of F4/80+ macrophages observed in the LSI groups or 20 m.o. old mice is likely due to infiltrated macrophages. It is necessary to analyze the types of infiltrated macrophages, such as M1 or M2, through co-staining with markers like CD206, CD163, and others. Additionally, co-staining of F4/80 with CD68, CD11b, and/or CD45 should be performed to confirm that these infiltrated cells are indeed macrophages.

Response: We appreciate the reviewer's thoughtful comments. As the reviewer's insightful observation, there are very few F4/80⁺ macrophages in the endplate of 3 m.o. old mice or control mice. So the macrophages are probably infiltrating the endplate in LSI mice or 20 m.o. old mice. To validate that these infiltrated cells are indeed macrophages, we conducted the co-immunostaining of F4/80 with CD68 or CD11b in the LSI surgery mice and aged mice. The results demonstrated that F4/80 was mostly co-localized with CD68 and CD11b in the sclerotic endplate, as shown in the Supplementary Figure 7a of the revised manuscript. This indicates that these infiltrated cells are macrophages.

Macrophage polarization refers to an estimate of macrophage activation at a given point in space and time. As one literature review states, the polarized state of macrophage is dynamic across time and there is no scientific basis to justify dualistic models of macrophage polarization. Due to the lack of clearly defined criteria to score phenotypes, the use of terms M1 and M2 remains somewhat controversial. Therefore, macrophage polarization appears complex, for example, in tumors, macrophage polarization is sometimes more M2-like but with M1-associated gene expression (PMID: 20570887, PMID: 26365184, PMID: 16269622, PMID: 24812208).

Despite its shortcomings, there is no denying that macrophage polarization remains a very important classification for research on macrophages. As suggested by the reviewer, to clarify the cell type of the infiltrated senescent macrophages, we conducted the co-immunostaining of P16 with M2 marker, CD206 or M1 marker, iNOS. The data revealed that the majority of the P16-positive cells expressed the M2 polarization marker CD206, as opposed to the M1 polarization marker iNOS in the endplates of LSI surgery mice and aged mice (supplementary figure 7b, c). This suggests that most senescent macrophages tend to exhibit the M2 markers. Similarly, recent research have shown that senescent macrophages express high levels of M2 markers (PMID: 26260587, PMID: 37267953). It has also been reported that although senescent macrophages differ from M1 and M2 expressing markers of macrophage polarization (PMID: 35247131). Together, based on our results, the macrophage is senescent in the degenerated endplate and exhibits a more M2-like marker.

b) It is not logical to assume that macrophages are already senescent upon infiltration into the endplate. Rather, macrophages may undergo senescence after infiltration. To address this question, the authors should analyze when macrophages infiltrate the endplate and when they start expressing senescence markers. This can be achieved by analyzing sequential time points, such as examining 6, 12, and 16 m.o. old mice in addition to the 3 m.o. and 20 m.o. old mice used in the aging models.

Response: We appreciate the reviewer's valuable suggestion. To investigate when senescent macrophages infiltrate the endplates, we conducted co-staining of P16 with F4/80 in the endplates from mice of different ages in Supplementary Figure 6 in the revised manuscript. The results showed that there were few P16⁺F4/80⁺ cells in the endplates of 3 m.o. old mice, while at 6 m.o. old mice, the infiltrated F4/80⁺ cells began to appear. Interestingly, we observed a high proportion of co-localization of P16 with F4/80 in the endplates of 6 m.o. old mice. Subsequently, the amount of P16⁺F4/80⁺ cells increased in the endplates of 12 m.o. old mice and peaked in the endplates of 16 m.o. old mice, while remaining at a high level in the endplates of 20 m.o. old mice. Together, these data indicate that macrophages are already senescent upon infiltration into the endplate.

c) Diagnosing macrophage senescence is not straightforward, as even non-senescent macrophages can express markers like p16 and SA- β -gal. Based on Figure 5 (E, F), around 40% of macrophages in the control (sham) mice are double positive for F4/80 and tdTomato (about 60% in the LSI group). These 40% of double positive macrophages found in the control young mice are unlikely to be senescent macrophages. Therefore, at most, 20% of macrophages in the LSI group could be considered "senescent". However, it remains uncertain whether these 20% of "senescent macrophages" are truly senescent and whether they are responsible for the elevation of IL-10 expression and the phenotypic changes observed in the LSI models. Thus, the

title of the manuscript, “Senescent macrophages mediates angiogenesis --- by IL-10”, is not justified at this stage.

Response: We appreciate the reviewer’s important comments. As the reviewer pointed out, although SA- β -gal and P16 were commonly used markers of cellular senescence, they could also be detected in non-senescent cells. Therefore, due to the lack of a specific marker, to determine whether macrophages undergo senescence in degenerated endplates, we comprehensively evaluate multiple markers, including SA- β -gal, P16, DNA damage markers (SADS, γ H2AX), and SASP (PMID: 34508069 PMID: 33328614). Therefore, we simultaneously tested multiple markers at the same time in macrophages to demonstrate the senescent phenotype.

Based on the histological results, there were few F4/80⁺tdTom⁺ cells in the endplates of the Sham group. We repeated the flow cytometry analysis to explore the potential reason for the detection of F4/80⁺tdTom⁺ cells in the Sham group. Against expectations, F4/80⁺tdTom⁺ cells were still detected in the Sham group. This may be due to the fact that in the flow cytometry experiments, it was difficult to ensure that there was no mixed nucleus pulposus or vertebral body tissue when separating the endplate tissue. In addition, during the preparation of single-cell suspensions of endplate tissue, the digestion time is usually longer because the endplate is composed of cartilage or bone tissue. Although we shortened the digestion time when repeating the experiment, it still inevitably affected the cell state, which may also contribute to the presence of a certain number of F4/80⁺tdTom⁺ cells in the sham group. However, the number of tdTom⁺F4/80⁺ cells increased significantly in the LSI group relative to the sham group as shown in Fig. 5, which is more important than the proportion of F4/80⁺tdTom⁺ cells over F4/80⁺ cells.

Regarding whether senescent macrophages are responsible for the elevation of IL-10, we found that senescent macrophages are an important source of IL-10 by co-immunofluorescent staining (IL-10, TdTom with F4/80) in vivo and ELISA in vitro. Additionally, neutralizing antibodies to IL-10 inhibited endplate degeneration in vivo and tube formation of endothelial cells induced by supernatant of senescent macrophages in vitro. To our best knowledge, the removal of P16 is considered as an important route for the clearance of senescent cells (PMID: 33758201, PMID: 35881544). In this study, we bred *lyz2-cre* mice with *Cdkn2a(P16)* flox/flox mice to specifically delete *Cdkn2a(P16)* in macrophages and found that both angiogenesis and degenerative phenotypes in endplates were significantly inhibited. Vivaly, co-immunofluorescence of IL-10 with F4/80 or P16 demonstrated that the number of double positive cells in the endplates of LSI group was significantly diminished in the conditional knockout mice (Supplementary Figure 16). Consistent with our findings, it has also been reported that senescent macrophages secrete high levels of IL-10 (PMID: 18279031).

d) Senescent cells typically secrete pro-inflammatory factors as part of their SASP, whereas IL-10 is an anti-inflammatory cytokine. Therefore, it is important to consider

the possibility of M2-like macrophages, rather than "senescent macrophages", being involved in the increase of IL-10 in the endplates of LSI and aging models. The subtypes of infiltrated macrophages (and/or resident macrophages) and cytokines including IL-10 should be verified through co-staining etc., as mentioned above.

Response: We appreciate the reviewer's valuable suggestion. As we mentioned above, macrophage polarization appears complex and dynamic across time. The senescent macrophages detected in the degenerated endplate expressed more M2 marker, CD206, as indicated in Supplementary Figure 7b of the revised manuscript. It is consistent with literature showing that senescent macrophages express high levels of M2 markers ((PMID: 26260587, PMID: 37267953). However, it has also been reported that although senescent macrophages express markers of macrophage polarization, they are dissimilar to M1 and M2 (PMID: 35247131). We found that senescent macrophages are an important source of IL-10 by co-immunofluorescent staining (IL-10, TdTom with F4/80) in vivo and ELISA in vitro (Figure 9a-c). Co-immunofluorescence of IL-10 with CD206 or iNOS showed that the cells expressing IL-10 mainly co-localized with CD206⁺ cells rather than iNOS⁺ cell (Supplementary Figure 15). Although IL-10 is usually considered to be secreted mainly by M2-type macrophages, it has also been reported that senescent macrophages secrete high levels of IL-10 (PMID: 18279031). Vitrally, co-immunofluorescence of IL-10 with F4/80 or P16 demonstrated that the number of double positive cells in the endplate of LSI group was significantly diminished in the *Cdkn2a^{Allyz2}* mice (Supplementary Figure 16). Therefore, the macrophage is senescent in the degenerated endplates and exhibits more M2-like marker and the senescent state of macrophages does not contradict the increased secretion of IL-10.

e) Furthermore, the authors state that "the most specific markers for cellular senescence are SADS and γ H2A.X" (page 8). However, this is an overstatement and incorrect. Additionally, in Figures 5H and 5J, SADS⁺/F4/80⁺ or γ H2A.X⁺/F4/80⁺ double positive cells should be normalized to the total number of F4/80⁺ cells rather than per area.

Response: We appreciate the reviewer's important suggestion. We apologize for the overstatement. We have rewritten this sentence as "We tested the sensitive molecular markers for tracking DNA damage, including senescence-associated distension of satellites (SADS) and γ H2A.X foci, which are characteristic features of senescent cells". There were few F4/80⁺ cells in the endplate of Sham group, while F4/80⁺ cells infiltrated the degenerated endplates of LSI surgery mice and aged mice. That is the reason that we calculate the number of SADS⁺/F4/80⁺ or γ H2A.X⁺/F4/80⁺ double positive cells per area. Additionally, we also calculated the percentage of the SADS⁺/F4/80⁺ or γ H2A.X⁺/F4/80⁺ double positive cells over F4/80⁺ cells in LSI surgery group or aged group in Supplementary Figure 8 in the revised manuscript.

2. The authors knocked out p16 in macrophages and granulocytes and observed that p16 knockout prevents endplate sclerosis in the LSI models. p16 depletion only will not reverse the state of senescence, therefore the striking phenotypic changes found in the KO mice are somewhat surprising. The authors should at least analyze the expression levels of cytokines, particularly IL-10, using Q-PCR and determine the cells expressing IL-10 by performing co-immunostaining of IL-10 with p16, gH2A.X, F4/80, CD206 etc to see if IL-10 acts as a downstream mediator of p16 in the development of sclerosis.

Response: We appreciate the reviewer's important comments. To our best knowledge, the removal of P16 is considered an important route for the clearance of senescent cells (PMID: 33758201, PMID: 35881544). Deletion of *Cdkn2a* (P16) in RANK⁺ cells efficiently prevents subchondral cellular senescence (PMID: 35881544). Inducible knockout of *Cdkn2a* (P16) in Cdh5⁺ endothelial cells inhibits endothelial senescence by glucocorticoids (PMID: 33758201). Similarly, our study also found that the deletion of *Cdkn2a* (P16) in Lyz2⁺ macrophages effectively prevented the cell senescence in the endplates.

We found that senescent macrophages are an important source of IL-10 by co-immunofluorescent staining (IL-10, TdTom with F4/80) and ELISA demonstrated that the level of IL-10 was elevated in the endplates of 20-month-old mice relative to 3-month-old mice (Figure 9a-c). Co-immunofluorescence of IL-10 with CD206 or iNOS showed that the cells expressing IL-10 mainly co-localized with CD206⁺ cells rather than iNOS⁺ cell (Supplementary Figure 15). Although IL-10 is usually considered to be secreted mainly by M2-type macrophages, it has also been reported that senescent macrophages secrete high levels of IL-10 (PMID: 18279031). Vitrally, co-immunofluorescence of IL-10 with F4/80 or P16 demonstrated that the number of double positive cells in the endplates of LSI group was significantly diminished in the *Cdkn2a^{ALyz2}* mice (Supplementary Figure 16). Therefore, the macrophage is senescent in the degenerated endplates and exhibits more M2-like marker and the senescent state of macrophages does not contradict the increased secretion of IL-10. We have included this in the discussion part of the revised manuscript.

3. The phenotypic changes observed in the iDTR system suggest the involvement of macrophages in sclerosis. However, it is crucial to analyze the expression changes of cytokines such as IL-10, IL-1 β , TNF in the macrophage/granulocyte-depleted endplates.

Response: We appreciate the reviewer's important comments. We conducted the qRT-PCR analysis to evaluate the expression changes of IL-10, IL-1 β , and TNF α in the endplate from *iDTR^{ALyz2}* mice at 8 weeks after LSI surgery with or without DT injection. The data showed that the level of IL-10, IL-1 β , and TNF α was inhibited in the endplates from *iDTR^{ALyz2}* mice by DT injection. We have included it in the Supplementary Figure 17 of the revised manuscript.

4. About the clearance of senescent cells using D+Q: It should be noted that current senolytic treatments, including D+Q, are not specific to senescent cells. To evaluate the

effect of senescent cells on sclerosis, additional senolytics that act through different mechanisms, such as ABT737, should be considered.

Response: We appreciate the reviewer's important comment. As the reviewer pointed out, since the first senolytic were discovered in 2015, many potential senolytics have been reported through different mechanisms, such as Kinase inhibitors, Bcl-2 family protein inhibitors, Naturally occurring polyphenols, Heat-shock protein inhibitors, and so on. Among them, Dasatinib (D) and Quercetin (Q) were first discovered senolytic and had been substantially investigated. D acts as Tyrosine Kinase inhibitor and Q is a natural polyphenol, which acts in part by inhibiting BCL-2 family members. ABT-737 is the inhibitor of BCL-2 family and is chemically similar to ABT263. The first human trials with D+Q showed that D+Q treatment could significantly reduce markers of senescence in various tissues. While, two major clinical side effects of ABT263, thrombocytopenia, and neutropenia were reported due to off-target effects (PMID: 37548098, PMID: 32686219). Together, we chose D+Q treatment in the present study.

As the reviewer stated, senolytic treatments are not specific. To further evaluate the effect of senescent macrophage on endplate sclerosis, we also included the macrophage knockout models (iDTR^{ΔLyz2} mice or CL treatment) and P16 conditionally knockout model (Cdkn2a^{ΔLyz2} mice). Comprehensively, we demonstrated the involvement of senescent macrophages in the degeneration of the endplates.

5. In most figures, the data heavily rely on histological analyses. Strengthening the data by including Western blotting and Q-PCR for proteins/mRNAs like CD31, Emcn, OSX, OCN, F4/80, among others, would provide additional support.

Response: We appreciate the reviewer's valuable suggestion. We included the western blotting and qRT-PCR analysis of F4/80, CD31, OSX, P16, and γ H2A.X in the LSI or sham surgery group with D+Q or DMSO treatment. The data demonstrated that LSI surgery significantly induced the expression of F4/80, CD31, OSX, P16, and γ H2A.X in the endplates. While the D+Q treatment prevented the LSI-induced increase of them. We have included these data in Supplementary Figure 3 of the revised manuscript.

6. TGF β has been reported to promote endplate sclerosis (Scientific Reports | 6:27093 | DOI: 10.1038/srep27093). TGF β is a SASP factor, and both IL-10 and TGF- β can be expressed by M2 macrophages. Therefore, it is essential to examine the expression levels of TGF β in the models investigated in this study.

Response: We appreciate the reviewer's important suggestion. The TGF β has been reported to be involved in endplate sclerosis (PMID: 27256073). As the reviewer's suggestion, we conducted the co-immunofluorescent staining of TGF β with F4/80 in vivo in the Supplementary Figure 18a of the revised manuscript. The data showed that the TGF β could co-localize with F4/80 in the endplates of the LSI group or aged group. Additionally, the western blot also showed that the total level of TGF β in the senescent macrophages increased in vitro relative to naive macrophage, as revealed in Supplementary Figure 18b, c of the revised manuscript. However, as the literature

reported, the latent TGF β was mainly stored in the extracellular matrix and then released and activated in responding to stimuli (PMID: 24487640). Similarly, the level of active TGF β increased in the endplates at 2W, 4W, and 8W post LSI surgery, but the total TGF β only increased in the endplates at 2W post LSI surgery and then returned to basal level at 4W and 8W post LSI surgery (PMID: 27256073). Therefore, the activation of TGF β is more important than the total TGF β . Together, these indicate that the TGF β released by senescent macrophages might participate in the endplate sclerosis, but may not be the most important source of active TGF β . We have included these in the discussion part of the revised manuscript.

Reviewer #4 (Remarks to the Author):

Fang and colleagues have examined the role of senescent macrophages in endplate sclerosis using multiple animal models and pharmacological approaches. They gathered data to suggest that inhibition of vegf or IL10, deletion of p16 expression in cells of the macrophage lineage, deletion of macrophages, or administration of the senolytics desatinib and quercetin all inhibit endplate sclerosis. From these results they conclude that IL10 produced by senescent macrophages contributes to endplate vascularization. This idea is novel but conceptually it is difficult understand.

This is because the presence of macrophages in chondrocytic tissues is dependent on blood vessel invasion. In addition, the amount of CD31+Emcn+ blood vessels present at 4 and 8 wk after LSI surgery or with aging is directly dependent on the size of the pores created by the osteoclasts, as previously shown by the authors. However, for unknown reasons, in this manuscript the authors do not provide any information about osteoclasts. Because osteoclasts are essential for the enlargement of the pores and mineralization of the endplates, it is likely that changes in osteoclasts are a critical determinant of the changes in porosity and sclerosis with the different interventions. Without the osteoclast component it is difficult to envision how the mechanism proposed by the authors works. Moreover, because the authors analyze the effects of their genetic or therapeutic intervention at time points when the endplate sclerosis is well developed, it is not possible to pin down the cascade of events that contribute to the effects measured.

Response: We appreciate the reviewer's important comment. Our previous study (Nat Commun. 2019;10(1):5643. doi:10.1038/s41467-019-13476-9) and the present work illustrate the presence of angiogenesis, osteoclast activity and osteogenesis during the degeneration of the endplates. Our data reveals that the LSI-induced increase in the angiogenesis (Figure 6) and the number of Trap⁺ osteoclasts (Supplementary Figure 11) are reduced in *Cdkn2a*^{ALyz2} mice. Considering that the endplate is composed of cartilaginous components from the perspective of development, the action of osteoclasts may also require the invasion of blood vessels. The term "angiogenesis osteogenesis coupling" has demonstrated the strong relationship between osteogenesis and angiogenesis (PMID: 31903130). This is confirmed by the fact that blocking

angiogenesis can hinder the degeneration of the endplates by the inhibition of vegf (Supplementary Figure 1). Together, we believe that the coupled actions of angiogenesis, osteoclast activity, and osteoblast activity lead to endplate sclerosis. When the angiogenesis is inhibited, the osteoclastic and osteogenic activities are also hindered. In the present study, we focused on the role of senescent macrophage-mediated angiogenesis in the osteogenesis and sclerosis process, so we did not detect osteoclasts. We apologized for the unclear statement and included the detection of osteoclasts in different experimental designs (Supplementary Figure 11, 14). Interestingly, the pattern of increased osteoclasts in the endplates with age (3, 6, 12, 16, and 20-month-old mice) is consistent with the trend of increased senescent macrophages, as indicated by the figure below and Supplementary Figure 6.

Another problem with the work presented is that even under normal physiological circumstances, β -galactosidase activity is enriched in particular cell types, such as mature tissue macrophages and osteoclasts. Moreover, in macrophages, high p16Ink4a expression is not directly associated with senescence. (OdgrenC Connect Tissue Res, 2006; Kopp et al. Histol Histopathol, 2007; Cudejko et al. Blood, 2011; Hall et al. Aging, 2017).

Response: We appreciate the reviewer's important comment. As stated by the reviewer, there is a lack of specificity in the detection of cellular senescence, and therefore a range of assays are needed to confirm the cellular senescence phenotype, including elevated SAbGal, P16 expression, DNA damage, secretion of SAS, and decline in cellular proliferative capacity. We comprehensively analyzed multiple assays before considering that macrophages undergo senescence during endplate degeneration. To further evaluate the role of cellular senescence in endplate degeneration, we employed senolytic, specific knockout of P16, and genetic or pharmacologic clearance of macrophages. Together, we suggest that senescent macrophages are involved in the angiogenic and sclerotic alterations of the degenerated endplates. We have included it in the discussion part of the revised manuscript.

In many instances there is not enough detail provided in the methods for the work to be reproduced and some of the assays are not described. Overall, the conclusions of this work are not well supported by the data provided. The following are a list of other concerns raised by the work presented.

Response: We appreciate the reviewer's comment. We have provided detailed information and included missing assays in the methods, as shown in the revised manuscript. We have addressed all the comments with additional experimentation and clarification. Detailed point-to-point responses to reviewers' critiques are described below.

Major points

1- The authors use several markers to indicate endplate sclerosis, such as porosity, endplate scores, area of CD31+Emcn+ blood vessels, and number of Osx and Ocn positive cells. However, these markers are redundant and their changes are correlated because the amount of vessels and osteoblastic cells depends on the number and size of the pores. Moreover, the endplate scores are not defined in the methods.

Response: We appreciate the reviewer's comment. As we mentioned above, endplate degeneration is typically characterized by porosity and sclerosis. It is the coordinated action of angiogenesis, osteoclasts, and osteoblasts that leads to these phenotypes in the endplates. The present work mainly focused on the exploration of the mechanisms of endplate sclerosis. We intend to indicate that the senescent macrophage-induced angiogenesis transports osteogenic precursor cells to mediate endplate sclerosis. So these markers, including endplate scores, area of CD31+Emcn+ blood vessels, and number of Osx and Ocn positive cells, were used to comprehensively analyze and testify our hypotheses.

2- The findings that CD31+EMCN+ blood vessels are present in the porous endplates after LSI surgery and aging (Fig. 1) has been previously shown by the authors.

Response: We appreciate the reviewer's comment. In our previous study, we only showed the phenotype that CD31+EMCN+ blood vessels are present in the endplates after LSI surgery and aging, but we did not conduct further investigation on the mechanisms of angiogenesis in the endplate. Here, we focused on the mechanisms of angiogenesis in the degenerated endplate. To round out the story, we also showed the angiogenesis of CD31+EMCN+ blood vessels in the degenerated endplate in the present study.

3- In figure 2A and H, the authors should show a picture of the whole intervertebral area so that the extent of the b-gal staining can be appreciated.

Response: We appreciate the reviewer's valuable suggestion. We have provided new representative SA β -gal images with whole intervertebral in the revised manuscript, as indicated in Figure 2a, h of the revised manuscript.

4- In Figure 2 D-E and K-L the comparison between HMGB1 in chondrocytes versus some unidentified cells in the pores is meaningless.

Response: We appreciate the reviewer's comment. As the most abundant chromatin related non histone binding protein in the nucleus, HMGB1 stabilizes nucleosomes and regulates the transcription of many genes in the nucleus. When cells undergo senescence, they lose nuclear HMGB1 and secrete it into the extracellular environment, where it acts as a pro-inflammatory cytokine. The release of HMGB1 has been proven to be a marker of cellular senescence. In Figure 2 d-e and k-l, we compared the expression of HMGB1 in the endplates between sham group and LSI group or young group and aged group. The data showed that HMGB1 mainly localized at the nucleus in the sham group and young group, while the HMGB1 was decreased and released out of the nucleus in the LSI group and aged group. Herein, these data assisted in proving the existence of cellular senescence in the degenerated endplate.

5- In figure 2 M-N, the fold change in expression most of the genes analyzed is about 4. This is statistically very improbable. The authors should confirm whether this is indeed the case and provide the gene expression without the normalization of the control values to 1 in these experiments and others throughout the manuscript.

Response: We appreciate the reviewer's comment. We have provided the gene expression without the normalization of the control values to 1 in these experiments as below:

	3M_1	3M_2	3M_3	3M_4	3M_5	3M_6	20M_1	20M_2	20M_3	20M_4	20M_5	20M_6
P16	24.40	24.82	24.20	24.03	23.40	23.78	22.00	21.97	21.35	20.90	21.07	21.05
P53	25.20	25.53	24.98	24.78	24.09	24.52	23.16	23.07	22.58	21.46	21.37	22.04
P21	25.12	25.29	24.85	24.40	23.86	24.22	23.16	22.78	22.24	21.57	21.42	22.16
IL-1 β	24.65	25.03	24.58	24.18	23.86	24.06	22.50	22.11	22.53	20.97	21.18	21.54
IL-6	25.60	25.96	25.47	25.09	24.75	24.82	24.18	24.89	24.33	23.68	23.56	23.81
Bactin	16.04	16.23	15.88	15.49	15.00	15.21	15.52	15.69	15.48	14.55	14.55	15.01

6- In some of the images the immunohistochemistry for Osx and OCN is overstained because all the cells are brown, including some of the chondrocytes seen in the high magnification images. In any case, the significance of the changes seen in the number of these cells is unclear, other than just a reflection of the cellular content of the pores. For example, the authors claim that administration of D+Q to aged mice (Fig 4J-K) decreases the number of Osx cells within the pores. How would this contribute to the decrease in porosity by D+Q? In contrast to these findings, D+Q increases osteoblastic cells in aged bone (Farr et al, Nature Medicine 2017).

Response: We appreciate the reviewer's valuable suggestion. We apologize for the inappropriate representative images and have replaced them in the revised manuscript, as indicated in Figure 1m, n. As we explained above, the coupled action of angiogenesis-osteoclasts-osteoblasts leads to endplate sclerosis. When the angiogenesis is inhibited, the osteoclastic and osteogenic activities are also hindered. In the present study, we showed that senescent macrophage-mediated angiogenesis was involved in endplate degeneration. The administration of D+Q could inhibit senescent macrophage-induced angiogenesis, then may decrease the transportation of osteogenic and osteoclastic precursor cells to the endplates. We have included it in the discussion part

of the revised manuscript.

7- In figure 5E-F, the quantification of Tomato positive macrophages indicates that 40% of macrophages are positive for p16 in sham operated mice. The origin of these macrophages is unclear because the histology images do not reveal the presence of macrophages in the endplates of sham operated mice. In any case, this is in agreement with multiple publications indicating that macrophages express p16 independent of senescence. However, in the other panels of figure 5 the authors quantified senescence markers in sham mice at sites where there are no macrophages present and therefore all readings in sham mice are zero. Thus, all these graphs and images merely indicate the presence or absence of macrophages. Does the deletion of p16 or D+Q decrease the presence of SADS or DNA damage markers in macrophages?

Response: We appreciate the reviewer's important comments. For histological sectioning and staining, we conducted coronal sections of the spinal tissues and chose sections in the middle part for staining and analysis. In the histological sections, we could clearly distinguish the boundary between the endplate tissue and the nucleus pulposus or vertebral body tissue. We repeated the flow cytometry analysis in an attempt to explore the potential reason for the detection of F4/80⁺ cells in the control group. However, against expectations, F4/80⁺ cells were still detected in the sham group in the flow cytometry data. This may be due to the fact that in the flow cytometry experiments, although we used a microscope to separate the endplate tissue, it was difficult to ensure that there was no mixed nucleus pulposus or vertebral body tissue. In addition, during the preparation of single-cell suspensions of endplate tissues, the digestion time was longer because the endplates were cartilage or bone tissues. Although we shortened the digestion time when repeating the experiment, it still inevitably affected the cell state, which may also contribute to the presence of a certain number of tdTom⁺F4/80⁺ cells in the sham group. However, it is more important that the number of tdTom⁺F4/80⁺ cells in the sham group was significantly lower than that in the LSI group, rather than the percentage of TdTom⁺F4/80⁺ cells over F4/80⁺ cells. Together, these limitations do not affect the results that the number of tdTom⁺F4/80⁺ cells increased significantly in the LSI group relative to sham group, as indicated by flow cytometry data.

As the reviewer's suggestion, we also conducted the SADS detection in macrophages in the endplates of P16 knockout mice or D+Q-treated mice. The data showed that the conditional knockout of P16 in Lyz2⁺ cells or D+Q treatment inhibited the presence of SADS in the F4/80⁺ macrophages in the LSI surgery mice, as indicated in Supplementary Figure 10.

8- Is there any potential reason for why the macrophages become senescent in the pores of mice with LSI surgery?

Response: We appreciate the reviewer's important comments. Macrophage populations and phenotypes can change with alterations in the surrounding microenvironment (Linehan and Fitzgerald, 2015). Meanwhile, aberrant mechanical loading has been

proven to induce cell senescence (Zhao J et al. Arthritis Res Ther. 2023 Apr 4;25(1):54. doi: 10.1186/s13075-023-03037-3. PMID: 37016437; PMCID: PMC10071751; Zhang H et al. Ann Rheum Dis. 2022;81(5):676-686. doi:10.1136/annrheumdis-2021-221513). In our study, both the LSI model and the aging model exhibited aberrant mechanical loading of the cartilage endplates, so we speculate that the cause of macrophage senescence may come from aberrant mechanical loading. Interestingly, we found that most of the macrophages that infiltrate into the cartilage endplate were already senescent, as indicated by immunofluorescence experiments of different age mice (Supplementary Figure 6). Exploring the specific causes and molecular mechanisms of macrophage senescence is a very valuable study. However, our research mainly focuses on the changes in downstream molecular mechanisms caused by senescent macrophages, further exploration can be conducted in the future.

9- In figure 6, the authors should quantify the efficacy of deletion of p16, even if in macrophages from the bone marrow. Do p16 deleted macrophages become less senescent and produce less IL10? Does p16 deletion alter osteoclast number?

Response: We appreciate the reviewer's valuable suggestion. The efficacy of P16 deletion was confirmed by co-immunostaining of P16 with F4/80. Moreover, we conducted the co-immunofluorescent staining of F4/80 with IL-10 or P16 with IL-10 in the endplate of *Cdkn2a^{ff}* mice or *Cdkn2a^{ALyz2}* mice after LSI surgery or sham surgery. The data demonstrated that the number of F4/80⁺IL-10⁺ cells and P16⁺IL-10⁺ cells decreased significantly in the endplates after LSI surgery in *Cdkn2a^{ALyz2}* mice relative to *Cdkn2a^{ff}* mice (Supplementary Figure 16). Additionally, we also conducted the SADS detection in macrophages in the endplates of P16 knockout mice. The data showed that the conditional knockout of P16 in *Lyz2⁺* cells inhibited the presence of SADS in the F4/80⁺ macrophages in the LSI surgery mice, as indicated in Supplementary Figure 10.

As the reviewer's suggestion, Trap staining were performed in the endplates of *Cdkn2a^{ff}* mice or *Cdkn2a^{ALyz2}* mice after LSI surgery or sham surgery. The data showed that the number of Trap⁺ osteoclasts decreased significantly in the endplates after LSI surgery in *Cdkn2a^{ALyz2}* mice relative to *Cdkn2a^{ff}* mice. We have included these data in the revised manuscript as Supplementary Figure 11.

10- It is unclear how 1 month of administration of D+Q to old mice, in which endplate sclerosis is established, can significantly decrease porosity and seemingly rejuvenate the cartilaginous portion. This is in stark contrast to the findings of Novais et al (Nat Commun 2021) showing that administration of D+Q to 18-month-old mice with establish intervertebral disk and endplate degeneration had no effect.

Response: We appreciate the reviewer's comments. After carefully reading Novais et al (Nat Commun 2021) research, we thought that the reasons for the differences in the experimental results are as follow: Our administration method is intragastric administration three times a week. While Novais et al conducted the intraperitoneal

injection once a week. More frequent administration and different routes of administration may cause differences in experimental results. To sum up, although there are some differences in the results of senolytics administration in the elderly mice, Novais et al's work and our study both confirmed that the treatment of eliminating senescent cells is beneficial for the recovery of the endplate.

11- In the experiments in which macrophages are deleted (Fig 7 and 8) the number of osteoclasts must be directly impacted, as macrophages are the precursors of osteoclasts. However, this is not even considered in the discussion of these experiments. The relevance of this work is unclear.

Response: We appreciate the reviewer's valuable suggestion. We conducted the Trap staining in the endplate with genetic or pharmacologic clearance of macrophages. The data showed that the deletion of macrophages in LSI surgery mice or aged mice decreased the number of Trap⁺ osteoclasts in the endplate. We have included it in the revised manuscript, as shown in Supplementary Figure 14. We have discussed this shortage in the revised manuscript.

12- The experiment of fig. 9A is not described in the methods or in the figure legend. The figure is not labeled appropriately.

Response: We appreciate the reviewer's valuable suggestion. We have included the detailed information of Fig. 9A in the methods and figure legend in the revised manuscript.

13- In Fig 10C, the majority of macrophages expresses Tomato. However, senescence is a cell fate that affects a small minority of cells in vivo. Thus, it is unlikely that these markers reflect cellular senescence. In the image provided there are also a number of cells other than macrophages that express IL10.

Response: We appreciate the reviewer's comment. As stated by the reviewer, there is a lack of specificity in the detection of cellular senescence, and therefore a range of assays are needed to confirm the cellular senescence phenotype, including elevated SAbGal, P16 expression, DNA damage, secretion of SASPs, and decline in cellular proliferative capacity. We comprehensively analyzed multiple assays before considering that macrophages undergo senescence during endplate degeneration. To further evaluate the role of cellular senescence in endplate degeneration, we employed senolytics, specific knockout of P16, and genetic or pharmacologic clearance of macrophages. The co-immunofluorescent staining of Il-10, F4/80 with tdTom in Figure 9c indicated that senescent macrophage is the main source of IL-10. However, cells other than senescent macrophages may also be a source of IL-10, such as bone marrow cells (PMID: 29249357). We have included it in the discussion part of the revised manuscript.

The background of the two panels in figure 10E is very different and therefore is difficult to appreciate any differences. In fig 10F, the expression of *cdkn2a* should be

measured.

Response: We appreciate the reviewer's valuable suggestion. We apologize for the improper representative images in Figure 9e and have replaced them in the revised manuscript. As the reviewer's comment, we also analyzed the expression of *cdkn2a* in Fig 9F in the revised manuscript.

Minor points

1- The labels of the Y axis of the graphs showing CD31+EMCN+ blood vessels are confusing because the graph shows %0 and the legends show percent.

Response: We appreciate the reviewer's comments. We feel sorry for the mistake. The Figure legends have been corrected in the revised manuscript.

2- The sex and background strain of the mice are not described.

Response: We appreciate the reviewer's comments. Only male mice were used for LSI model and Aging model in this study. The spontaneous osteoporosis that occurs in old female mice could affect the subchondral bone, which is also an important characteristic in cartilage pathology. To avoid this, we use male mice in our study, according to other similar studies. We have described the background strain of the mice in detail in the method section.

3- Many measurements are reported per area (/Ar) but these should be clarified.

Response: We appreciate the reviewer's valuable suggestion. The area is obtained by measuring the targeted endplate using image J software. We have added this explanation to the method section.

4- There are many typos trough out the manuscript.

Response: We appreciate the reviewer's valuable comments. As suggested by the reviewer, we have revised these issues.

REVIEWER COMMENTS

Reviewer #1 (Remarks to the Author):

No further questions

Reviewer #2 (Remarks to the Author):

Authors have addressed all my comments. I have no further comments.

Reviewer #3 (Remarks to the Author):

The manuscript is improved by this revision. However, the following questions/comments should be addressed.

1. It is well known that senescent cells and macrophages share many characteristics, and there is still debate on whether macrophages senesce or whether there is a clear definition for macrophage senescence. Thus, the term “senescent macrophages” used in the current manuscript is too definite. Changing “senescent macrophages” to “senescent-like macrophages” in the title and text is desirable and will be more easily accepted by readers.
2. It has been repeatedly reported that acute injury itself can induce senescent cell accumulation at the injured site. However, few senescent cells can be found from naturally aged mice in general. The authors showed that senescent-like macrophages were already senescent upon recruitment to the endplate both in LSI and naturally aged mouse models. The route/mechanism of senescent-like macrophage recruitment/accumulation in LSI and naturally aged mouse models may be different. The authors should at least provide commentary on it in the Discussion section.
3. The Figs were labeled by upper cases, but lower cases were used in the text and legends.
4. Result section: page 14, line 309. “pSTA3” should be “pSTAT3”.

5. Discussion section: Please check page 17, line 387, if “MAPK, SIRT, mTOR” should be “AMPK, SIRT, mTOR”.

Reviewer #4 (Remarks to the Author):

The authors have addressed most of the concerns raised. Nonetheless, some issues related to the new findings and text in this revised version, should be addressed:

A-Based on the analysis of mice at 3, 6, 12, and 20 month-of-age presented in Supplementary Fig. 6, the authors state that (Line 163) a high proportion of co-localization of P16 with F4/80 was seen in the endplates of 6-month-old mice. In the images shown, there are no p16 negative F4/80 cells, all cells are yellow. This is in agreement with the evidence that p16 is expressed in most macrophages, independent of senescence. Thus, in the absence of other markers such as SADS and gH2AX foci, the authors should avoid calling p16+ macrophages as senescent, and tone down the conclusions based on these findings.

B-The description of these new findings (presented in Supplementary Fig. 6) is not accurate. Specifically, the authors claim that “the amount of P16+F4/80+ 164 cells increased in the endplates of 12-month-old mice and peaked in the endplates of 16-month-old mice, while remaining at a high level in the endplates of 20-month-old mice”. However, the quantification presented shows no obvious differences between 12, 16, and 20 months. The description of this data should be corrected.

C-The description of the SADS method is still lacking detail, for example what was the minimum number of macrophages counted, within each condition? What kind of microscope was used?

D-Line 205. “...the number of Trap+ osteoclasts decreased significantly in the endplates after LSI surgery in Cdkn2a Δ Lyz2 mice relative to Cdkn2af/f mice” should be replaced by “the number of Trap+ osteoclasts after LSI surgery was lower in the endplates of Cdkn2a Δ Lyz2 mice relative to Cdkn2af/f mice”

E-Line 214. The use of “additionally” and “also” is a repetition, please use one or the other.

F-Line 379. “The senolytic we used to eliminate senescent cells by inhibiting the anti-apoptotic proteins”. This sentence needs to be fixed.

G-Line 380. “We conducted the western blot of the anti-apoptotic Bcl-2 protein and pro-apoptotic protein Bax on the senescent BMDMs. The results demonstrated that the senescence in macrophages is accompanied by an increase in anti-apoptotic activity”.

Western blot does not assess protein activity, this sentence should be fixed.

H-The new paragraphs on the discussion about TGF β and macrophage polarization should be condensed.

I-It is unclear the relationship of the whole paragraph spanning from line 383 to line 394 to the work presented in the manuscript. This should be removed.

J-The title and text should be thoroughly reviewed for grammar and typos.

REVIEWER COMMENTS

Reviewer #1 (Remarks to the Author):

No further questions

Response: We appreciate the reviewer's efforts to review our manuscript.

Reviewer #2 (Remarks to the Author):

Authors have addressed all my comments. I have no further comments.

Response: We are encouraged by the reviewer's comments. We appreciate the reviewer's efforts to review our manuscript.

Reviewer #3 (Remarks to the Author):

The manuscript is improved by this revision. However, the following questions/comments should be addressed.

1. It is well known that senescent cells and macrophages share many characteristics, and there is still debate on whether macrophages senesce or whether there is a clear definition for macrophage senescence. Thus, the term "senescent macrophages" used in the current manuscript is too definite. Changing "senescent macrophages" to "senescent-like macrophages" in the title and text is desirable and will be more easily accepted by readers.

Response: We are encouraged by the reviewer's comments. Thank you for your careful review and valuable suggestions on our manuscript. We have replaced the term "senescent macrophages" with "senescent-like macrophages" both in the title and the main text. We believe that this modification will enhance the acceptance of our paper among a wider readership.

2. It has been repeatedly reported that acute injury itself can induce senescent cell accumulation at the injured site. However, few senescent cells can be found from naturally aged mice in general. The authors showed that senescent-like macrophages were already senescent upon recruitment to the endplate both in LSI and naturally aged mouse models. The route/mechanism of senescent-like macrophage recruitment/accumulation in LSI and naturally aged mouse models may be different. The authors should at least provide commentary on it in the Discussion section.

Response: We appreciate the reviewer's valuable comments. Acute injury-induced cellular senescence is a specific physiological process, which has the characteristics of clear senescent trigger, short-term senescent signal, and rapid senescent cell clearance. Different from acute senescence, chronic senescence does not have a specific program, but it is a random process. Multiple and persistent stresses that act on tissues and organs may induce chronic senescence (Yang L, Wang B, Guo F, et al. FFAR4 improves the senescence of tubular epithelial cells by AMPK/Sirt3 signaling in acute kidney injury. *Signal Transduct Target Ther.* 2022;7(1):384. Published 2022 Nov 30. doi:10.1038/s41392-022-01254-x; Lin X, Jin H, Chai Y, Shou S. Cellular senescence and acute kidney injury. *Pediatr Nephrol.* 2022;37(12):3009-3018. doi:10.1007/s00467-022-05532-2).

In our study, the Lumbar Spine Instability (LSI) model was established by removing the L3–L5 spinous processes, the supraspinous ligaments, and the interspinous ligaments. This model did not directly affect the endplates but induced the degeneration of the endplate through unstable mechanical loading. Therefore, the LSI model is preferring to the mechanism of accelerating chronic senescence. In the naturally aged mouse model, multiple and persistent stresses are the main causes

of chronic senescence. In both models, persistent factors such as aberrant mechanical loading, oxidative stress, cytotoxicity, and mitochondrial dysfunction may be the potential mechanisms of recruitment of senescent-like macrophages. However, the specific mechanisms still need to be further explored in the future. We have included these in the discussion part of the revised manuscript.

3. The Figs were labeled by upper cases, but lower cases were used in the text and legends.

Response: We appreciate the reviewer's valuable suggestion. In the revised version of the manuscript, we have changed the labels of the figures to lowercase. This adjustment will enhance the clarity and accuracy of our presentation.

4. Result section: page 14, line 309. "pSTA3" should be "pSTAT3".

Response: We apologize for the typo. We have corrected it in the revised manuscript.

5. Discussion section: Please check page 17, line 387, if "MAPK, SIRT, mTOR" should be "AMPK, SIRT, mTOR".

Response: We appreciate the reviewer's comment. We feel sorry for our carelessness. We have corrected it in the revised manuscript.

Reviewer #4 (Remarks to the Author):

The authors have addressed most of the concerns raised. Nonetheless, some issues related to the new findings and text in this revised version, should be addressed:

A-Based on the analysis of mice at 3, 6, 12, and 20 month-of-age presented in Supplementary Fig. 6, the authors state that (Line 163) a high proportion of co-localization of P16 with F4/80 was seen in the endplates of 6-month-old mice. In the images shown, there are no p16 negative F4/80 cells, all cells are yellow. This is in agreement with the evidence that p16 is expressed in most macrophages, independent of senescence. Thus, in the absence of other markers such as SADS and gH2AX foci, the authors should avoid calling p16+ macrophages as senescent, and tone down the conclusions based on these findings.

Response: We are encouraged by the reviewer's comments. Based on our experimental results, with the increase of age, p16/F4/80 double positive cells gradually infiltrate into the endplate. In this section, we are sorry for the imprecise statement and modify the statement of senescence to p16 high macrophages. Additionally, to enhance the precision, we have revised the term "senescent macrophage" to "senescent-like macrophages" in the title and main text of the revised manuscript.

B-The description of these new findings (presented in Supplementary Fig. 6) is not accurate. Specifically, the authors claim that "the amount of P16+F4/80+ cells increased in the endplates of 12-month-old mice and peaked in the endplates of 16-month-old mice, while remaining at a high level in the endplates of 20-month-old mice". However, the quantification presented shows no obvious differences between 12, 16, and 20 months. The description of this data should be corrected.

Response: We appreciate your insightful comments. In the revised manuscript, we have revised the statement as "the number of P16⁺F4/80⁺ cells increased in the endplates of 12-month-old mice and

remained at a high level in the endplates of 16, and 20-month-old mice”.

C-The description of the SADS method is still lacking detail, for example what was the minimum number of macrophages counted, within each condition? What kind of microscope was used?

Response: We are sorry for the lack of detail on SADS methodology. We have further described the specific experimental details and equipment in the methodology.

D-Line 205. “...the number of Trap⁺ osteoclasts decreased significantly in the endplates after LSI surgery in Cdkn2a^{ΔLyz2} mice relative to Cdkn2a^{fl/fl} mice” should be replaced by “the number of Trap⁺ osteoclasts after LSI surgery was lower in the endplates of Cdkn2a^{ΔLyz2} mice relative to Cdkn2a^{fl/fl} mice”

Response: We sincerely appreciate the reviewer for the careful comment. As suggested by the reviewer, we have corrected “the number of Trap⁺ osteoclasts decreased significantly in the endplates after LSI surgery in Cdkn2a^{ΔLyz2} mice relative to Cdkn2a^{fl/fl} mice” to “the number of Trap⁺ osteoclasts after LSI surgery was lower in the endplates of Cdkn2a^{ΔLyz2} mice relative to Cdkn2a^{fl/fl} mice”.

E-Line 214. The use of “additionally” and “also” is a repetition, please use one or the other.

Response: We appreciate the reviewer’s comment. We have deleted “also” in the revised manuscript.

F-Line 379. “The senolytic we used to eliminate senescent cells by inhibiting the anti-apoptotic proteins”. This sentence needs to be fixed.

Response: Thank you for pointing this out. As suggested by the reviewer, we have corrected the “The senolytic we used to eliminate senescent cells by inhibiting the anti-apoptotic proteins” into “We used senolytic to eliminate senescent cells by inhibiting anti-apoptotic proteins”.

G-Line 380. “We conducted the western blot of the anti-apoptotic Bcl-2 protein and pro-apoptotic protein Bax on the senescent BMDMs. The results demonstrated that the senescence in macrophages is accompanied by an increase in anti-apoptotic activity”. Western blot does not assess protein activity, this sentence should be fixed.

Response: We appreciate your insightful comments. We have corrected this sentence as “The results demonstrated that the senescence in macrophages is accompanied by an increase in the expression level of the anti-apoptotic Bcl-2 protein”.

H-The new paragraphs on the discussion about TGFβ and macrophage polarization should be condensed.

Response: We appreciate your insightful comments. We have condensed this section in the revised manuscript to make the paragraphs more concise.

I-It is unclear the relationship of the whole paragraph spanning from line 383 to line 394 to the work presented in the manuscript. This should be removed.

Response: We appreciate your insightful comment. This section was added to respond to another reviewer's comment on the relationship of mitochondrial dysfunction with cellular senescence.

J-The title and text should be thoroughly reviewed for grammar and typos.

Response: We sincerely appreciate the reviewer's careful reading. In our resubmitted manuscript, the typo is revised. Thanks for your correction.

REVIEWERS' COMMENTS

Reviewer #3 (Remarks to the Author):

The authors addressed my concerns.

Reviewer #4 (Remarks to the Author):

The authors have successfully addressed all the remaining comments.

REVIEWER COMMENTS

Reviewer #3 (Remarks to the Author):

The authors addressed my concerns.

Response: We are encouraged by the reviewer's comments. We appreciate the reviewer's efforts to review our manuscript.

Reviewer #4 (Remarks to the Author):

The authors have successfully addressed all the remaining comments.

Response: We appreciate the reviewer's efforts to review our manuscript.